# Why is cyclic dominance so rare?

Hye Jin Park[1,2], Yuriy Pichugin[1], Arne Traulsen[1]*

[1]Department of Evolutionary Theory, Max Planck Institute for Evolutionary Biology, Plön, Germany; [2]Asia Pacific Center for Theoretical Physics, Pohang, Republic of Korea

**Abstract** Natural populations can contain multiple types of coexisting individuals. How does natural selection maintain such diversity within and across populations? A popular theoretical basis for the maintenance of diversity is cyclic dominance, illustrated by the rock-paper-scissor game. However, it appears difficult to find cyclic dominance in nature. Why is this the case? Focusing on continuously produced novel mutations, we theoretically addressed the rareness of cyclic dominance. We developed a model of an evolving population and studied the formation of cyclic dominance. Our results showed that the chance for cyclic dominance to emerge is lower when the newly introduced type is similar to existing types compared to the introduction of an unrelated type. This suggests that cyclic dominance is more likely to evolve through the assembly of unrelated types whereas it rarely evolves within a community of similar types.

## Introduction

Natural populations ranging from microbial communities to animal societies consist of many different individuals. Some individuals compete with each other to exploit a shared resource (*Hardin, 1960*; *Connell, 1983*), whereas others coexist (*Morris et al., 2013*). Interactions affect the death or reproduction of individuals and thus shape the composition of populations (*Friedman and Gore, 2017*). Different types of individuals are distinguishable at the interaction level and they have a complex interaction structure (*Farahpour et al., 2018*). Because interaction structures themselves can support the coexistence of multiple types, they have been extensively studied in ecology and evolution (*Gross et al., 2009*; *Allesina and Levine, 2011*). A particularly exciting type of interaction is cyclic dominance, in which each type dominates another one but is in turn dominated by a separate type, leading to a Rock-Paper-Scissors cycle (*Maynard Smith, 1982*; *Hofbauer et al., 1998*; *Szab and Fth, 2007*) as sketched in *Figure 1A*. None of the types fixates in the population, because each type is dominated by one type while it simultaneously dominates a third type. Thus, it has been argued that this type of interaction can support biodiversity (*Reichenbach et al., 2007*). Cyclic dominance has therefore attracted substantial attention and it has been extensively studied theoretically (*Maynard Smith, 1982*; *Hofbauer et al., 1998*; *Frean and Abraham, 2001*; *Hauert et al., 2002*; *Reichenbach et al., 2007*; *Szab and Fth, 2007*; *Mathiesen et al., 2011*; *Jiang et al., 2011*; *Allesina and Levine, 2011*; *Szolnoki et al., 2014*).

A famous example of this type of cyclic dominance in biology is toxin production in *Escherichia coli* (*Kerr et al., 2002*; *Kirkup and Riley, 2004*; *Cascales et al., 2007*). Toxin-producing (or colicinogenic) *E. coli* cells can purge cells that are sensitive to the toxin. However such toxin producers are dominated by resistant cells that do not produce the toxin. Once common, resistant cells are again dominated by sensitive cells, which avoid the costs of resistance. This leads to cyclic dominance in a Rock-Paper-Scissors manner, as shown in *Figure 1B*. Another example is the mating strategies of North American side-blotched lizards *Uta stansburiana* (*Sinervo and Lively, 1996*). The strategy of males guarding several females dominates the strategy of males guarding only a single female. However, sneaky strategies under which males secretly mate with guarded females can become dominant over the strategy of males guarding several females. Once such a sneaky strategy is common,

*For correspondence:
traulsen@evolbio.mpg.de

**Competing interests:** The authors declare that no competing interests exist.

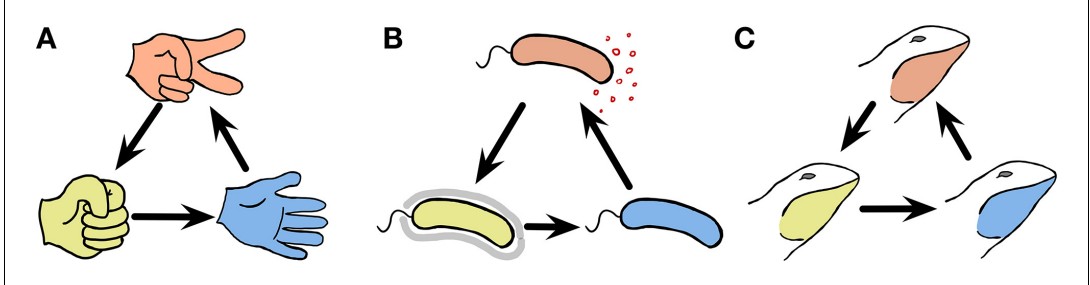

**Figure 1.** Cyclic dominance triplets across the scale of organisms. In cyclic dominance, each type dominates one type and it is in turn dominated by another type. An arrow points from the dominated toward the dominant type. (**A**) The three actions in the game, Rock-Paper-Scissors cyclically dominate each other. (**B**) This game can also describe bacterial interactions (*Kerr et al., 2002*; *Kirkup and Riley, 2004*; *Cascales et al., 2007*): Some *E. coli* cells (orange) can produce a toxin that suppresses the survival of sensitive cells (blue). Hence toxin-producing cells (orange) dominate sensitive cells, whereas they are dominated by resistant cells (yellow). However in the absence of toxin-producing cells, the sensitive cells dominate resistant cells, exhibiting cyclic dominance. (**C**) Such dynamics can also occur in higher animals, as typified by the mating strategies of male side-blotched lizards (*Sinervo and Lively, 1996*; *Sinervo et al., 2007*): Strategies under which individual lizards guard many females (orange) can be invaded by a sneaker strategy that steals matings (yellow). If such a sneaker strategy is frequent, guarding a single mating partner (blue) can lead to higher mating success. However, once sneakers become rare again, guarding many females is beneficial, leading to cyclic dominance.

the strategy of guarding only a single female can be successful again leading to cyclic dominance among the mating types as illustrated in *Figure 1C*. In addition to *E. coli* and side-blotched lizards, other examples of cyclic dominance have been described in ecology: *Stylopoma spongites* (*Jackson and Buss, 1975*), *Drosophila melanogaster* (*Clark et al., 2000*), European lizards *Lacerta vivipara* (*Sinervo et al., 2007*), and plant systems (*Taylor, 1990*; *Lankau and Strauss, 2007*; *Cameron et al., 2009*). These types of cyclic dominance arise because of competition, which can happen within and between species at the same trophical level. Mating or sperm competitions are the basis of cyclic dominance within species observed in *U. stansburiana* and *D. melanogaster*, whereas common resource competition can happen both between and within species.

However, traditional theoretical work assumes a set of predefined cyclic dominance types without asking how they developed or came together. In ecosystems, the introduction of a new species through migration can lead to such cyclic dominance. In this context, it is often termed intransitive competition (*Allesina and Levine, 2011*; *Soliveres et al., 2015*; *Gallien et al., 2017*). However, immigrating species can also disturb and destroy cyclic dominance. In evolving populations, new types can arise through mutation and recombination. In the same manner, mutation and recombination can lead to the formation of cyclic dominance but can also lead to types that do not fit into such types of dominance and break the cycle. A recent experimental study (*Higgins et al., 2017*) indicated that in the assembly of microbial ecosystems based on 20 bacterial strains found in a single grain of soil, only 3 of almost 1000 triplets exhibited cyclic dominance. Other soil bacterial species (*Wright and Vetsigian, 2016*; *Friedman et al., 2017*) also displayed a lack of cyclic dominance. This rareness is found in both soil bacteria and plant systems (*Taylor, 1990*). Why is it so difficult for cyclic dominance to emerge by assembly or evolution? In this study, we ask the following question theoretically: How frequent is cyclic dominance in situations in which new types constantly arise, providing an opportunity for new cycles but also breaking old cycles at the same time?

Forming a cyclic dominance from a single type is challenging because as soon as the second type arises, the dominance type will take over the whole population, driving extinction of the other type. Therefore, a third type must arise before the population loses either of the two previous types. Such a precise timing of the arrival of a new type is critical for developing cyclic dominance and it can occur when new types arise at a high frequency, either through high mutation rates, recombination, or immigration (*Kotil and Vetsigian, 2018*; *Tarnita, 2018*). This rapid evolution can be achieved through both high mutation rates per capita and large population sizes (*Lanfear et al., 2014*; *Hague and Routman, 2016*; *Vahdati et al., 2017*; *Kotil and Vetsigian, 2018*; *Tarnita, 2018*). Thus, we considered a model in which the population naturally evolves to a large population size (*Park et al., 2019*), which allows the development of cyclic dominances via an evolutionary process.

Once we introduced our model in more detail, we will show that the interaction of ecology and evolution leads to increasing population size. This increases the chances that cyclic dominance arises, but it remains rare. Next, we rationalize this finding: While the lifespans of cyclic and non-cyclic dominance triplets are similar, it is more difficult to form cyclic dominance compared to non-cyclic dominance. The underlying reason is that similarity between parental and offspring payoffs suppresses the formation of cyclic dominance. Finally, we discuss which genealogical structure can promote or suppress cyclic dominance.

## Materials and methods

Interactions between individuals affect their death or birth. A traditional model for describing an interacting population is the generalized Lotka-Volterra equation (*Ginzburg et al., 1988*; *Bomze, 1995*; *Yoshida, 2003*). In particular, some studies (*Biancalani et al., 2015*; *Huang et al., 2015*; *Shtilerman et al., 2015*; *Barbier et al., 2018*; *Farahpour et al., 2018*) assumed that the interaction determines the likelihood of death from a pairwise competition. These interaction parameters can be written as a form of a matrix, including self-interaction. However, only a few studies considered novel mutations (*Huang et al., 2012*; *Shtilerman et al., 2015*; *Farahpour et al., 2018*). Drawing new interaction parameters for a new type and extending the interaction matrix, we considered such a novel mutation process. In addition, the size of the interaction matrix can be reduced when types go extinct. We traced an evolving population by dealing with this dynamically changing matrix.

We built the model based on individual reaction rules

$$
\begin{aligned}
I &\rightarrow I+I && \text{birth without mutation at rate } \lambda_b(1-\mu), \\
I &\rightarrow I+I' && \text{birth with mutation at rate } \lambda_b\mu, \\
I &\rightarrow \emptyset && \text{background death at rate } \lambda_d, \\
I+J &\rightarrow J && \text{death due to competition at rate } d_{ij},
\end{aligned}
\tag{1}
$$

where $I$ and $J$ are individuals of types $i$ and $j$, respectively. We assumed that all types are in the same trophic level, and thus there is only competition and no predation. All types have the same background birth and death rates. Only competition makes a difference (*Huang et al., 2015*; *Farahpour et al., 2018*; *Park et al., 2019*). Because the population always collapses when $\lambda_b \leq \lambda_d$, we only focused on $\lambda_b > \lambda_d$. For the sake of simplicity, we only considered well-mixed populations without any other high-order interactions.

Formulating the competition death rate $d_{ij}$ as a function of the payoff $A_{ij}$, we connected evolutionary game theory to the competitive Lotka-Volterra type dynamics (*Huang et al., 2012*; *Huang et al., 2015*; *Park and Traulsen, 2017*; *Park et al., 2019*; *Sidhom and Galla, 2020*). Note that $A_{ij}$ is the payoff of an individual of type $i$ from the interaction with an individual of type $j$. Because lower payoffs should increase the probability of death, we used an exponentially decaying function for the competition death rate as follows:

$$
d_{ij} = \alpha + e^{-A_{ij}},
\tag{2}
$$

where $\alpha > 0$ is the baseline death rate from competition, which ensures that the population remains bounded regardless of the value of the evolving payoffs $A_{ij}$. A larger payoff implies a lower death rate from competition. The overall competition death rate is always positive, such that we remain in the regime of the competitive Lotka-Volterra equations.

For a large population size, the abundance $x_i$ of type $i$ can be described using the competitive Lotka-Volterra equation

$$
\frac{d}{dT}x_i = (\lambda_b - \lambda_d)x_i - \sum_{j=1}^{n} d_{ij}x_ix_j,
\tag{3}
$$

where $n$ is the number of types in the population, used as a diversity index, and $T$ is the time. The stability of the population is determined by *Equation (3)*. In parallel, the stability between only two types can be determined by the two associated equations in *Equation (3)*, which are described by four payoff values of the two types.

Once a new mutant type arises during reproduction, new interactions occur. To describe these new interactions, we draw new payoff values from the parental payoff with Gaussian noise

$$
\begin{aligned}
A_{i'j} &= A_{ij} + \xi, \\
A_{ji'} &= A_{ji} + \xi, \\
A_{i'i'} &= A_{ii} + \xi,
\end{aligned}
\tag{4}
$$

where $\xi$ is a random variable sampled from a Gaussian distribution with zero mean and variance $\sigma^2$. This inheritance of payoffs with noise implies that the mutant type $i'$ slightly deviates from the parental type $i$. Here, we treat self-interaction $A_{ii}$ and interaction with other types $A_{ij(\neq i)}$ in the same way. Because of new interactions, the population composition changes over time, as shown in *Figure 2A*. We let the population evolve from a single type with a randomly drawn initial payoff from the normal distribution with mean $\ln(1000)$ and standard deviation 1. As a natural consequence of evolving payoffs, the average population size also evolves. Because different types are fully described by the payoff matrix, we can trace the evolving population by tracking the payoff matrix, as shown in *Figure 2B*. We do not consider any tradeoff: having higher payoffs does not cost anything. Hence, the evolution tends to increase the payoffs constantly and thus drives the system into a regime where payoff differences become smaller.

To construct a pairwise interaction network, we used the stability between two types. Hence the term interaction refers to the pairwise relationship, considering the stability between two types. There are four possible scenarios for stability (see Appendix 1):

- Dominance of type $i$:
  represented by $(i)\leftarrow\leftarrow(j)$ for $A_{ii}\gtrsim A_{ji}$ and $A_{ij}\gtrsim A_{jj}$.
- Dominance of type $j$:
  represented by $(i)\rightarrow\rightarrow(j)$ for $A_{ii}\lesssim A_{ji}$ and $A_{ij}\lesssim A_{jj}$.
- Bistability:
  represented by $(i)\leftarrow\rightarrow(j)$ for $A_{ii}\gtrsim A_{ji}$ and $A_{ij}\lesssim A_{jj}$.
- Coexistence:
  represented by $(i)\rightarrow\leftarrow(j)$ for $A_{ii}\lesssim A_{ji}$ and $A_{ij}\gtrsim A_{jj}$.

Constructing the interaction network, we can examine the formation and the collapse of cyclic dominance, as shown in *Figure 2C*. If the links are drawn from a random matrix, each stability scenario described above occurs with the same probability. Thus, we find a proportion of 0.50 dominance links and a proportion of 0.25 proportions for bistability and coexistence links, respectively. Because the networks can contain three different link types (dominance, bistability, and coexistence), both cyclic dominance and other types of triplets can be found. However, in the main text, we only focused on cyclic and non-cyclic dominance triplets which are composed of only dominance because the proportions of each triplet strongly depend on the proportions of link types. Hence at a given link composition (three dominance links), we investigate how often we can observe cyclic dominance compared to non-cyclic one.

## Results

### Evolution leads to increasing population size

The population dynamics described in *Equation (1)* appears simple, but its tracing is complicated because of the novel mutations. Due to the emergence of a new mutant and its consequences, the payoff matrix dynamically changes. As large payoffs lower competition death rates, types with higher payoffs are more likely to survive. Therefore, payoffs evolve to larger values, which increases the population size (*Park et al., 2019*). Since there is no tradeoff on the payoffs, the average payoff increases monotonically. This makes types become more similar, enhancing diversity (*Scheffer and van Nes, 2006*). However, the population size saturates at a certain level because of the baseline death rate $\alpha$ corresponding to resource limitation and enters a steady state (see Appendix 2).

For small $\alpha$ values (rich environments) in particular, the population size $N$ at the steady state becomes large, containing many different types (see *Figure 3A*). This evolution toward a large population induces rapid mutation. Once the populations size becomes large, new mutant types are generated faster than in smaller populations given a fixed mutation rate per individual. In this rapid

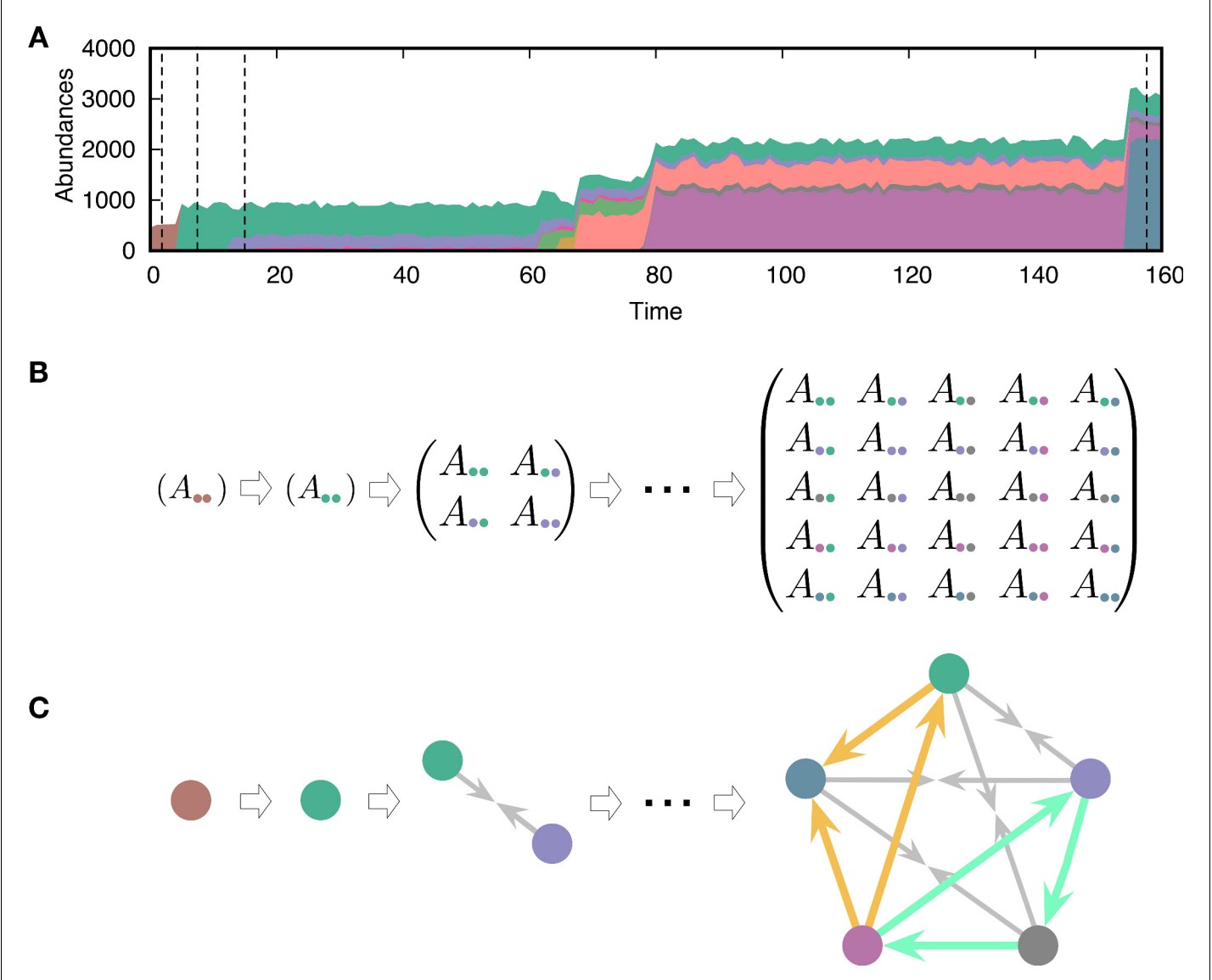

**Figure 2.** Evolving population dynamics and tracing its interactions and constructing a network. (**A**) Sample simulation of population dynamics over time. Different colors correspond to different types. The colored area represents the abundance of each type. Time $t$ is measured as the number of mutation events that occurred. (**B**) Interaction matrices between types at four different time points are marked by vertical dashed lines in panel **A**. Whenever a mutant emerges in the population, the diversity $n$ increases and the payoff matrix becomes larger. Extinction of resident types can also happen because of the new mutant, reducing the size of the matrix. For example, one of the first mutant types (green) dominates the resident type (brown) and takes over the entire population. (**C**) Interaction structures inferred from the interaction matrices. There are three possible relationships: dominance (with two different directions indicated by an arrow from the dominated type to the dominant type), coexistence (arrows from each type to the middle), and bistability (arrows towards both types, not present here). We focused on triplets as basic substructures of the network. There are two triplets composed of three dominance links, but they have different topologies. One of them is cyclic dominance (highlighted in green), and the other is non-cyclic dominance (highlighted in yellow).

mutation regime, a new mutant can arise before the population equilibrates, thereby establishing a cyclic dominance from a timely emerged mutant (*Kotil and Vetsigian, 2018*). Thus, cyclic dominance can be established when the populations enters the rapid mutation regime, as shown in *Figure 3B*. The proportions of triplets were averaged over all surviving realizations. In principle, we can observe a triplet from $t = 2$ even though there were fewer than three types on average. Because of the smaller average diversity $\langle n \rangle$, there were large fluctuations in measuring the proportions of triplets in the early regime ($t \lesssim 100$). However, the measurement became more accurate as diversity increased.

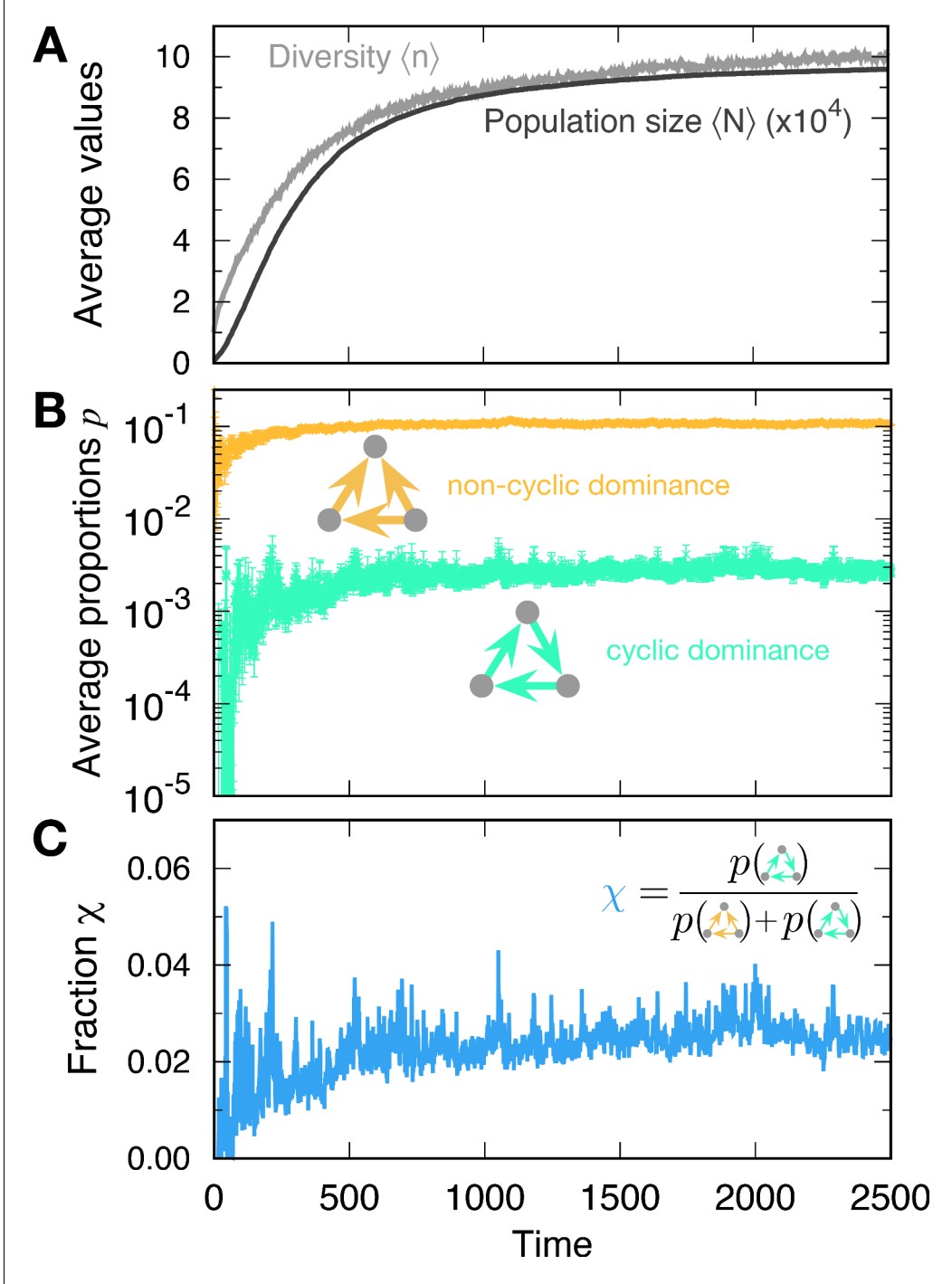

**Figure 3.** Formation of cyclic and non-cyclic dominance in the rapid mutation regime. (A) For a low baseline death rate $\alpha$, the population size $N$ tends to increases with new mutations. The average population size $\langle N \rangle$ and diversity $\langle n \rangle$ increase. When the population size $N$ becomes large, cyclic and non-cyclic dominance can emerge. (B) Cyclic dominance triplets were less abundant than non-cyclic ones. In the long run, the proportions saturated at around 0.0036 and 0.105 for cyclic and non-cyclic dominance, respectively. (C) To quantify the rarity of cyclic dominance compared with non-cyclic dominance, we calculate the fraction $\chi$ of cyclic dominance. In the early dynamics, it fluctuates because only a few realizations can form triplets because of the low average diversity. However, when large diversity is reached, the fraction became more stable, fluctuating around 0.033. This value is much smaller than the expected value when link types are randomly drawn ($\chi = 0.25$). indicating the rareness of cyclic dominance produced by novel mutations. (Simulation details: $\lambda_b = 0.9$, $\lambda_d = 0.4$, $\alpha = 5 \cdot 10^{-6}$, $\sigma = 1$, and $\mu = 10^{-5}$. Unless we mentioned the parameter values, the same parameters were used for the following figures as well. The average is based on 4889 samples that did not go extinct among 5000 realizations. In each time point, the ensemble average in B and C is performed only for $n \geq 3$ at a given time.).

## Cyclic dominance triplets are rare

The proportions of cyclic and non-cyclic dominance triplets increase in the early dynamics and quickly saturate. Whereas population dynamics illustrates the formation of both cyclic and non-cyclic dominance, cyclic dominance is much rarer than non-cyclic dominance. To quantify this rareness of cyclic dominance, we measured the fraction $\chi$, which is defined by the fraction of cyclic dominance triplets among all dominance triplets (cyclic and non-cyclic) (see *Figure 3C*). In steady state, the fraction yielded $\chi \approx 0.033$, indicating that one cyclic dominance triplet can be found among 30 dominance-composed triplets. If the pairwise relationships are random (all four possible links appear in the same probability 1/4, called a random network), then the fraction $\chi$ of cyclic dominance should be 0.25 because there are only two configurations of cyclic dominance triplets, whereas six configurations produce non-cyclic dominance triplets. Hence cyclic dominance that developed from our population dynamics is much rarer than expected from a random choice of interactions. To elucidate why it is the case, we checked how long each triplet can be sustained in the population and how often they emerge. First, we focused on the lifetime of cyclic and non-cyclic dominance and then moved on the formation of each triplet. When it comes to the chance to emerge cyclic and non-cyclic dominance, we argued that the genealogy structure shaped by eco-evolutionary dynamics will enhance or suppress the formation of cyclic dominance.

## The lifespans of cyclic and non-cyclic dominance triplets are similar

The rareness of cyclic dominance triplets may be caused by their shorter lifespan compared with that of non-cyclic dominance triplets. Thus, we investigated the lifespan of triplets first to understand the rareness of cyclic dominance. Once triplets arise in populations, we can identify them, and trace how long they persist. Lifespan distributions in the steady state of both cyclic and non-cyclic dominance triplets decayed algebraically. We plotted the complementary cumulative distribution functions (CCDFs), clearly revealing a power law decay, as shown in *Figure 4A*. Surprisingly, there was no difference in the lifespan of both triplets. Both cyclic and non-cyclic dominance triplets were destroyed in five mutation events on average. The non-cyclic dominance triplet has a higher chance of persisting longer, although the difference is small. In addition, the median is the same for both distributions because almost all probabilities are concentrated on short lifespans. In conclusion, lifespan does not explain why cyclic dominance is rarer than non-cyclic dominance. Hence, the lower chance for cyclic dominance to emerge is the reason.

## The condition to form cyclic dominance is more strict than that for non-cyclic dominance

Why is it more difficult for cyclic dominance to emerge than for non-cyclic dominance? One factor is that the conditions needed for an interaction matrix to provide cyclic dominance are more restrictive than those for non-cyclic dominance. For a matrix to reveal cyclic dominance, it is necessary that in each of the three columns, the three payoffs $A_{ij}$, $A_{jj}$, and $A_{kj}$ are ordered ($A_{ij}<A_{jj}<A_{kj}$ or $A_{ij}>A_{jj}>A_{kj}$). Conversely, the formation of non-cyclic dominance requires this condition to be satisfied only in a single column, whereas the other two columns should satisfy a less restrictive condition ($A_{ij}<A_{jj}$ and $A_{kj}<A_{jj}$ or vice versa). For example, in random payoff matrices where all payoffs are randomly drawn from the standard normal distribution, the fraction of cyclic dominance is $1/13 \approx 0.077$ (as shown in Appendix 4), which is smaller than the value of $2/8 = 0.25$ expected in a random network of directed links. This is because in the matrix approach, unlike a random network, links in a triplet are interdependent; self-interaction payoffs contribute to the character of several links at once.

## Similarity between parental and offspring payoffs suppresses the formation of cyclic dominance

The fraction $\chi \approx 0.077$ in the random matrix is still larger than that obtained from our population dynamics $\chi \approx 0.033$, implying there are other factors suppressing the development of cyclic dominance. A key reason is the correlation between payoffs. In our model, the elements of the payoff matrix are not fully independent because of heredity. Offsprings payoffs are derived from their parents payoffs. We found that this correlation between payoffs plays an important role in suppressing the formation of cyclic dominance. For example, let us imagine two different uncorrelated pre-existing types represented by a $2 \times 2$ random matrix. All elements are drawn from the standard

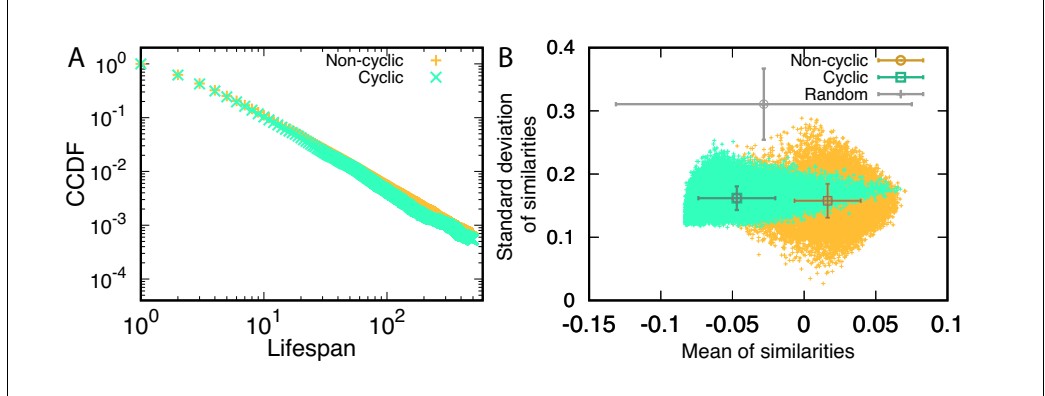

**Figure 4.** Properties of cyclic and non-cyclic dominance triplets. (**A**) We traced the identified cyclic and non-cyclic dominance triplets and measured their lifespan in steady state, $9500 \leq t \leq 10000$. The distributions of lifespan decayed algebraically, exhibiting a power law in the complementary cumulative distribution functions (CCDFs). The CCDF($x$) shows the probability to have a lifespan longer than $x$. Both distributions were almost identical, but the tail component of non-cyclic dominance was slightly heavier. (**B**) We defined a trait vector of each type to characterize cyclic and non-cyclic dominances. A triplet consists of three vectors, and a set of scalar products can be calculated for all pairs. Scalar products are calculated after all payoffs are shifted and normalized so that the average payoffs becomes zero and the norms are one. A positive scalar product indicates that two types are similar. We used the mean and standard deviation of these similarities to characterize a triplet and draw a scatter plot. The symbols with error bars indicate the mean and standard deviation of quantities in each axis. For comparison, we performed the same calculation for random matrices. For both cyclic and non-cyclic triplets, the standard deviations of similarities are much smaller than that of random matrices. However the means of similarities for both triplets show different behavior. Types were less similar to each other in cyclic dominance compared to random matrices, while non-cyclic dominance has more similar types compared to random matrice. For this analysis, we used 1944 surviving realizations and $5 \cdot 10^6$ sets of random matrices.

normal distribution. If a third type emerges from a pre-existing one by a mutation, the average fraction of cyclic dominance becomes $\chi \approx 0.02$ (the standard normal distribution is used for new payoffs and we average across $5 \cdot 10^5$ samples), which is lower than that of the assembly of thee uncorrelated types. This is because the payoffs of a new offspring are similar to its parental payoffs: the offspring is likely to have the same relationships with other types in a population as the parental type and any type dominated by the parent will also be dominated by the offspring. The triplets including these offspring and parental types are more likely to form non-cyclic dominance than a cyclic one. Hence the correlation between payoffs affects the fractions by which cyclic dominance emerges compared with non-cyclic dominance.

To check the effect of the correlation on emerging triplets, we measured the similarity between types as a proxy of the correlation between payoffs. We defined the trait vector $\vec{T}_l$ of the type $l$ using the row capturing with its own payoff and the column of the others payoff against it in the payoff matrix, $\vec{T}_l = (A_{l,i}, A_{l,j}, A_{l,k}, A_{i,l}, A_{j,l}, A_{k,l})$, similarly to *Farahpour et al., 2018*. Because the average payoff increases over time, we shifted all elements in those vectors by a constant to ensure that the average of all values is zero. Then, using normalized vectors after shifting we calculated the scalar product for all pairs of trait vectors as a similarity measure. Larger values indicate that the two types have more similar payoff values. Each triplet has three trait vectors and thus has three similarity measures. Taking the mean and standard deviation of these three similarities, we found that in cyclic dominance the three types tend to be less similar compared to non-cyclic dominance, as shown in *Figure 4B*. The inheritance plays a key role in the emergence of cyclic and non-cyclic triplets, giving rise to a correlation between payoffs. Because the payoff correlation is determined by a genealogy structure, in the following we investigated which of these structures have higher or lower chances to promote the emergence of cyclic dominance.

## Genealogical structure can promote or suppress cyclic dominance

Between the last common ancestor and the present types, there are intermediary types accumulating mutations between them. Genealogies tell us who is whose parent, tracing back to the common ancestor of the observed types. From the genealogy, we can infer how many mutations were accumulated by each type and the time at which they diverged. If two types have only accumulated a

few mutations from the most recent common ancestor, their payoffs are likely to be similar. Hence, the genealogy structure shapes the correlation between payoffs and affects the value of fraction $\chi$. To study the role of genealogies, we first characterized genealogies for three types based on accumulated mutational distances, see *Figure 5AB*. We found that for a triplet of types, the distribution of payoff elements could be characterized by five parameters of the genealogy, which all measure the number of mutations between types $w_1$, $w_2$, $x$, $y$, and $z$ (see Appendix 5).

We analyzed those mutational distances for cyclic and non-cyclic triplets found in the simulation at the steady-state. From the analysis, we inferred that the crucial parameter influencing the fraction $\chi$ of cyclic dominance is the fraction $F_l$ of mutations accumulated after each type evolves independently from others (the time $t_d$ in *Figure 5B*).

Payoff correlations must be developed before lineages become independent at $t_d$, depending on the mutational history ($w_1$ and $x$). With more mutations being accumulated after the types divergence, the correlation between types becomes weaker, increasing the chance of the cyclic dominance to emerge, see *Figure 5C*. On the other hand, with only few mutations after $t_d$ the strong correlation between parental and offsprings payoffs remains, which mostly induces the emergence of non-cyclic dominance. This means that the mutations accumulated after all three types diverge are important for decoupling their payoff correlations, increasing the chance to form the cyclic dominance. This finding agrees well with numerically investigated genealogies that gave the minimal and maximal values of the fraction $\chi$, called minimizer and maximizer genealogies, respectively. For minimizer genealogies, the majority of mutations were accumulated before $t_d$ (see Appendix 6). Conversely, for maximizer genealogies, the majority of mutations were accumulated after all three lineages diverged from each other.

In addition, numerically calculated minimizer genealogies almost completely suppressed the emergence of cyclic dominance $\chi_{\min} < 0.001$. For the maximizer, the fraction could be as high as $\chi_{\max} = 1/6 \approx 0.167$. Both the fraction of cyclic dominance arising from the random matrix $\chi \approx 0.077$ and that found in our simulations $\chi \approx 0.033$ fell between the minimal and the maximal values possible with the genealogy structure. The fractions $\chi$ from population dynamics are closer to the minimal value, as shown in *Figure 6*. Therefore, we can infer that in the genealogies occurred in population

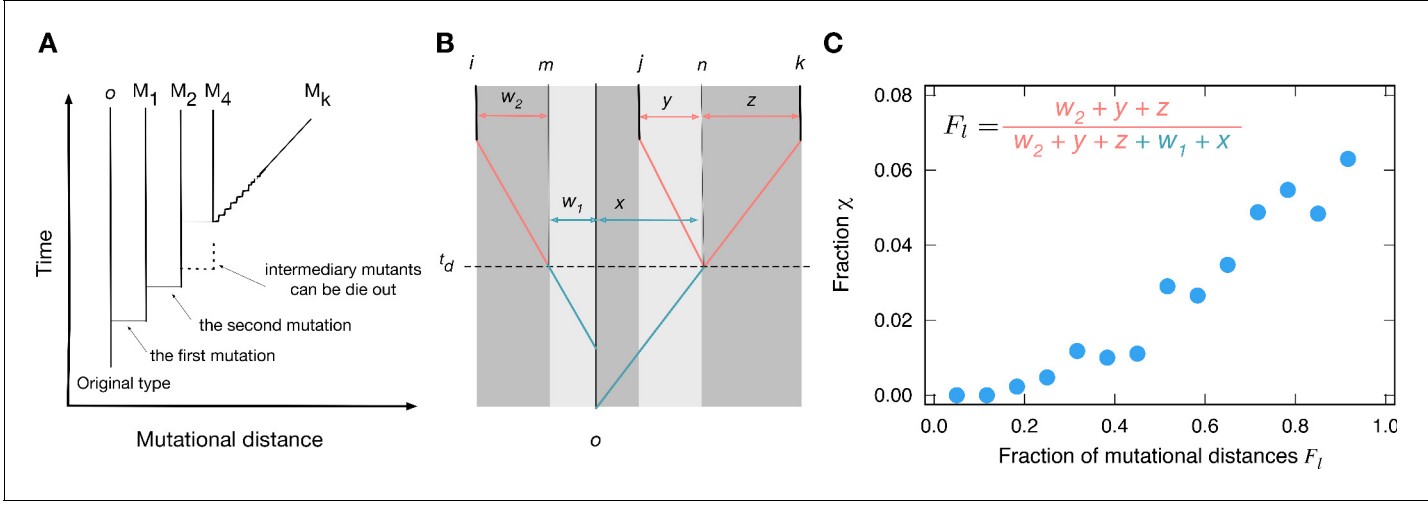

**Figure 5.** Genealogies and their effect on the fraction $\chi$ of cyclic dominance. (**A**). The scheme of genealogy. Vertical axis represents time, and horizontal axis represents mutational distance. Each solid vertical line corresponds to a single type. Horizontal lines indicate when mutation happens. Intermediary mutants can die out and a sequence of intermediary mutations is represented by a diagonal line. (**B**). An example of a genealogy for three types. Types $i$, $j$, and $k$ are the types under consideration, and type $o$ is their last common ancestor. Type $n$ is the last common ancestor of $j$ and $k$, and type $m$ is the ancestor of the type $i$ which existed at the moment when type $j$ diverges from type $k$. Each of the three types has its own independent lineage from time $t_d$ on, and the fraction of accumulated mutations before and after this time determined the chance for cyclic dominance to emerge. For a list of all other possible genealogies, see Appendix 5. (**C**) For the sake of simplicity, we defined $F_l$ as the fraction of accumulated mutations after time $t_d$ compared to all mutations since the last common ancestor, $F_l = \frac{w_2+y+z}{w_2+y+z+w_1+x}$. At the steady-state, when more mutations are accumulated after $t_d$ (large $F_l$ values), cyclic dominance can emerge more often by reducing the payoff correlations.

dynamics, the similarity of the new types to their parental type prevents the emergence of cyclic dominance.

## Discussion

Cyclic dominance is extremely interesting from a conceptual and theoretical perspective and it has thus been analyzed in great detail in mathematical biology (*Hofbauer et al., 1998*; *Hofbauer and Sigmund, 2003*; *Szab and Fth, 2007*). However, the theoretical literature typically refers to only a handful of examples in nature. Moreover, recent experiments have revealed that it is difficult for cyclic dominance to emerge in microbial populations (*Wright and Vetsigian, 2016*; *Friedman et al., 2017*; *Higgins et al., 2017*). Why is the establishment of cyclic dominance so difficult? To address this question, we used an evolutionary process with evolving interactions for the formation of such cyclic dominance instead of following the more conventional approach of using a predefined set of interactions. For example, *Kotil and Vetsigian, 2018* observed the formation of cyclic dominance in the fast evolution regime with adaptation, but the involved traits were predefined. However, we argue that it is difficult for cyclic dominance to emerge even in the presence of rapid evolution. In addition, our investigation shows no correlation between diversity and the probability to find cyclic

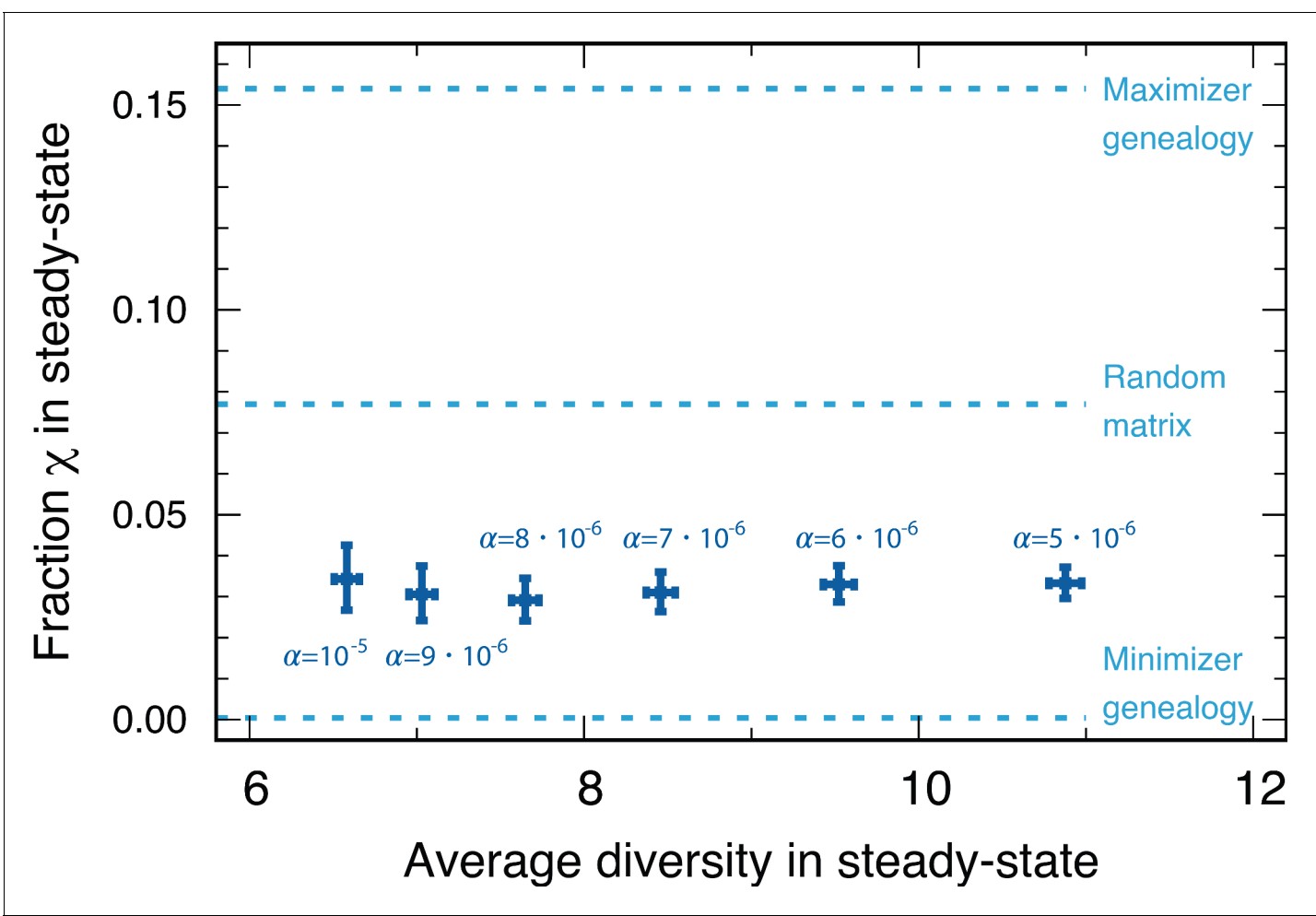

**Figure 6.** We compared the fractions $\chi$ of cyclic dominance in the steady state for various baseline death rates $\alpha$ from our simulations. We also denoted three reference fractions: the maximizer genealogy ($\chi \approx 0.167$), random matrix ($\chi \approx 0.077$), and minimizer genealogy ($\chi < 0.001$). For various $\alpha$, the fractions of cyclic dominance were similar, although the average diversities in steady state were different. The majority of fractions $\chi$ are between the one found for the minimizer geneaology and the fraction from the random payoff matrix. This implies that in the genealogies shaped by our population dynamics, the surviving offspring type is typically similar to the parental type. Each point is averaged over the surviving samples among 5000 realizations (4897, 4908, 4903, 4886, 4901, and 4889 samples for $\alpha = 10^{-5}, 9 \cdot 10^{-6}, 8 \cdot 10^{-6}, 7 \cdot 10^{-6}, 6 \cdot 10^{-6}, 5 \cdot 10^{-6}$, respectively).

dominance. The diversity observed in our model does not originate from the cyclic dominance structure, but from the generation on a time scale that is fast compared to equilibration of the system. Our results indicate cyclic dominance in general could be a mechanism to support diversity, but it is most probably not essential in a situation where it can emerge, as such situations are characterized by a high diversity in the first place. Even if we rescale the payoffs to prevent going towards neutrality which increases diversity, cyclic dominance remains rare as long as the cyclic dominance is not a main driver for the diversity. This result is robust even when we consider the population frequency weights, see Appendix 7. In our model, the non-cyclic dominance can reach a majority of the population, while the cyclic dominance typically remains bounded to a much lower frequency.

We also examined the circumstances under which cyclic dominance can appear more frequently. While the probability of assembling such an interaction structure by chance in a payoff matrix with uncorrelated random entries is small, the probability to evolve such an interaction structure is even smaller. The inheritance of interactions from parent to offspring is a key mechanism shaping the correlations between payoffs and determines the formation of cyclic dominance. From our approach, we found that the introduction of an uncorrelated type is crucial for the formation of cyclic dominance triplets. Because the migration of new species can be interpreted as such an introduction, our results suggest that cyclic dominance might be more frequent on an inter-species basis than on an intra-species basis. As widespread intransitive competition is found in ecological systems (*Soliveres et al., 2015*; *Gallien et al., 2017*), our manuscript nicely supports the basic idea that assembly of unrelated types is more likely to lead to cyclic triplets than evolution, in which emerging types are closely related.

Our approach, which reduces the complexity from continuous values to a categorical classification, may help to bridge the model dynamics and experimental data more easily. Experimental work has provided data regarding both the constituents of a microbial community but also the interactions between them. However for large communities, parameterizing all interactions in the model numerically makes it difficult to identify the fundamental factors shaping the dynamics. Reducing the complexity may permit study of the large scales of experimental data connecting the underlying model dynamics and large datasets.

An important limitation of our work is the assumption of global interactions. In our model, all individuals can interact with each other, ignoring the spatial population structure. A spatial model could localize the interactions and lead to the more frequent formation of cyclic dominance. Such a localization can foster cyclic dominance for a predefined cyclic set (*Durrett and Levin, 1997*; *Durrett and Levin, 1998*; *Frean and Abraham, 2001*; *Reichenbach et al., 2007*; *Szab and Fth, 2007*; *Mathiesen et al., 2011*; *Jiang et al., 2011*; *Mitarai et al., 2012*; *Szolnoki et al., 2014*; *Kelsic et al., 2015*; *Sneppen, 2017*). However, before moving into spatial models it appears essential to investigate this issue in the absence of all potentially confounding factors. Such models appear necessary for explaining why cyclic dominance within one species is not found often in nature, and they may open a new direction for the extensive theoretical work on this topic.

## Acknowledgements

HJP, YP, and AT thank the Max Planck Society for generous funding. HJP was also supported by (1) an appointment to the JRG Program at the APCTP through the Science and Technology Promotion Fund and Lottery Fund of the Korean Government and by (2) the Korean Local Governments - Gyeongsangbuk-do Province and Pohang city.

## Additional information

### Funding

| Funder | Author |
|---|---|
| JRG Program at the APCTP through Science and Technology Promotion Fund and Lottery Fund of the Korean Government and Korean Local Government Gyeongsangbuk-do Province and Pohang City | Hye Jin Park |

The funders had no role in study design, data collection and interpretation, or the decision to submit the work for publication.

### Author contributions

Hye Jin Park, Conceptualization, Data curation, Software, Formal analysis, Visualization, Writing - original draft, Writing - review and editing; Yuriy Pichugin, Formal analysis, Investigation, Writing - review and editing; Arne Traulsen, Conceptualization, Supervision, Writing - review and editing

### Author ORCIDs

Hye Jin Park (iD) https://orcid.org/0000-0003-3552-6275
Yuriy Pichugin (iD) https://orcid.org/0000-0003-3078-2499
Arne Traulsen (iD) https://orcid.org/0000-0002-0669-5267

### Decision letter and Author response

Decision letter https://doi.org/10.7554/eLife.57857.sa1
Author response https://doi.org/10.7554/eLife.57857.sa2

## Additional files

### Supplementary files

• Transparent reporting form

### Data availability

Our simulation code is available at https://github.com/Park-HyeJin/CyclicDominance (copy archived at https://github.com/elifesciences-publications/CyclicDominance).

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

## Appendix 1

### Pairwise relationship based on stability

For large population sizes, the change of abundances of types can be described by deterministic equations. Linear stability analysis reveals which types will persist in equilibrium (*Strogatz, 2000*). Using this stability, we determine the relationship between two types $i$ and $j$. In this section, we perform linear stability analysis first and then determine the pairwise relationships based on that stability.

To determine the stability between two types, type 1 and type 2, we focus on the two equations

$$\begin{aligned} \dot{x}_1 &= \lambda x_1 - d_{11}x_1^2 - d_{12}x_1x_2, \\ \dot{x}_2 &= \lambda x_2 - d_{21}x_1x_2 - d_{22}x_2^2, \end{aligned} \tag{A1.1}$$

where $\lambda = \lambda_b - \lambda_d$. Since the death rates $d_{ij}$ are all positive, the population would always go extinct for $\lambda < 0$. Thus, we only consider a positive $\lambda$. There are four fixed points $(x_1^*, x_2^*)$: One is the extinction of both types $(0,0)$, two are indicating each single-type population $(\lambda/d_{11}, 0)$ and $(0, \lambda/d_{22})$, and the fourth one is the coexistence point $\left(\frac{(d_{22}-d_{12})\lambda}{d_{11}d_{22}-d_{12}d_{21}}, \frac{(d_{11}-d_{21})\lambda}{d_{11}d_{22}-d_{12}d_{21}}\right)$. To check the stability of each fixed point, we calculate the Jacobian matrix $\mathbf{J}$,

$$\mathbf{J} = \begin{pmatrix} \lambda - 2d_{11}x_1 - d_{12}x_2 & -d_{12}x_1 \\ -d_{21}x_2 & \lambda - d_{21}x_1 - 2d_{22}x_2 \end{pmatrix}, \tag{A1.2}$$

and then calculate the eigenvalues for the fixed point.

The extinction point $(0,0)$ is always unstable, because both eigenvalues are identical to $\lambda > 0$. On the other hand, the three other fixed points have one negative eigenvalue $-\lambda$, indicating that they are saddle or stable fixed points; If the second eigenvalue is positive, the fixed point becomes a saddle and is unstable. If the second eigenvalue is negative, the fixed point is stable. Hence the fixed point $(\lambda/d_{11}, 0)$ becomes stable when $d_{21} > d_{11}$, because the eigenvalue is $(d_{11} - d_{21})\lambda/d_{11}$. In the same manner, the condition $d_{12} > d_{22}$ can be obtained for a stable fixed point $(\lambda/d_{11}, 0)$. Lastly, the coexistence point has an eigenvalue $-\frac{(d_{11}-d_{21})(d_{22}-d_{12})}{d_{11}d_{22}-d_{12}d_{21}}\lambda$. For meanginful values of abundances, stable coexistence points have to satisfy the conditions $x_1^* > 0$ and $x_2^* > 0$. Together with these restrictions, we can find the coexistence fixed point is stable only if both $d_{11} > d_{21}$ and $d_{22} > d_{12}$ are satisfied.

We classify the pairwise relationship based on these stabilities. Dominance relationships are given if only a single-type fixed point is stable while all other fixed points are unstable. When both single-type fixed points are stable, a bistability relationship is drawn. A coexistence relationship is achieved when only the coexistence 1xed point is stable. We summarized the stabilities of 1xed points at a given condition and named the pairwise relationship in *Appendix 1—table 1*.

**Appendix 1—table 1.** Given conditions, the stabilities of four fixed points are shown with its corresponding relationship between two types.

Stable fixed points are marked by S, while unstable fixed points are marke by U. Extinction is always unstable, while the stabilities of other fixed points depend on conditions. For the condition in the first row, $d_{11} < d_{21}$ and $d_{12} < d_{22}$, type 1 can survive in the equilibrium, indicating the dominance of type 1. The second condition indicates the dominance of type 2. When both single-type fixed points are stable, the population can end up either type 1 or 2 population showing bistability. For the last condition, only the coexistence fixed point is stable.

| Conditions | Fixed points | | | | Relationship |
|---|---|---|---|---|---|
| | $(0,0)$ | $(x_1^*, 0)$ | $(0, x_2^*)$ | $(x_1^*, x_2^*)$ | |
| $d_{11} < d_{21}$ and $d_{12} < d_{22}$ | U | S | U | U | Dominance of type 1 |
| $d_{11} > d_{21}$ and $d_{12} > d_{22}$ | U | U | S | U | Dominance of type 2 |
| $d_{11} < d_{21}$ and $d_{12} > d_{22}$ | U | S | S | U | Bistability |
| $d_{11} > d_{21}$ and $d_{12} < d_{22}$ | U | U | U | S | Coexistence |

## Appendix 2

### Population size and diversity at steady-state for various baseline death rates

In this section, we show how the population size $N$ and diversity $n$ evolve in time $t$ with various baseline death rates $\alpha$. Since larger payoffs imply lower death rates, individuals with larger payoffs tend to survive. As a result of evolution, the overall death rates decrease, which enlarges the population size. This can be interpreted that species improve their efficiency to consume a resource. However, if resources are limited, the population size is confined. We implement this resource limit by introducing the baseline death rate $\alpha$. The death rates from the competition cannot be lower than the baseline death rate, and thus the population size saturates at a certain level at the end, $\lambda/\alpha$. For various $\alpha$, we run 5000 independent stochastic simulations and measure the average population size and diversity in time $t$, see *Appendix 2—figure 1A and B*. The average runs over the surviving samples at a given time $t = 10000$, and we denote the averaged quantity $O$ as $\langle O \rangle$. As we expected, the population sizes evolve to $\lambda/\alpha$, and thus larger $\alpha$ leads to smaller population sizes.

Large population size reduces the time until new types occur in the population, and at the same time it takes longer to equilibrate. Thus, a large population size increases the number of different types in the population via two processes: (i) Even when the most competitive type is expected to fixate in the population, new types which are less fit can constantly emerge due to mutation. (ii) A second mechanism is ecological tunneling (*Kotil and Vetsigian, 2018*): Even though the population dynamics yields the extinction of certain types, it can be rescued by the emergence of a new type which supports the coexistence of types predicted to be extinct. From those effects, larger populations have higher diversity *Appendix 2—figure 1B*. We compare the average population size and diversity in the stationary regime as well, see *Appendix 2—figure 1C*.

We also take a closer look at the average payoff values. Because of selection, the average payoff increases in time, but the increment becomes smaller as the average increases, see *Appendix 2—figure 1D*. After a transient time, the average payoffs increases logarithmically. It means that the natural selection induced by the payoff difference becomes weaker and is almost neural at the end. Hence, in the long run, the diversity mainly originates from the mutation-selection balance.

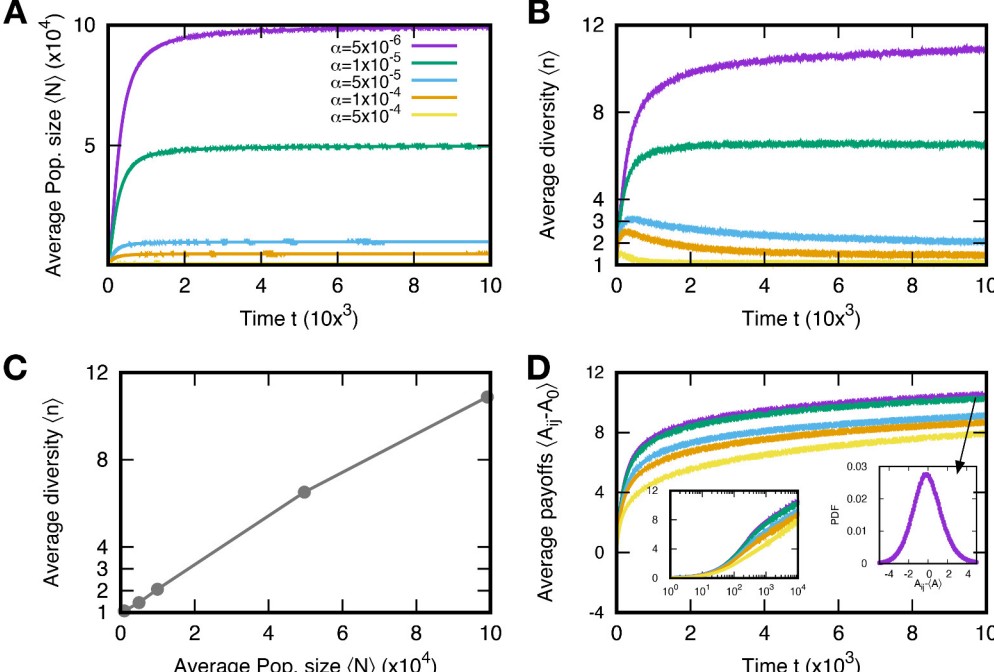

**Appendix 2—figure 1.** Population size and diversity for various baseline death rates $\alpha$. The average population size $\langle N \rangle$ and diversity $\langle n \rangle$ in mutation event time $t$ are shown in (**A**) and (**B**), respectively. We run 5000 independent simulations and use surviving samples at $t = 10000$ to obtain average quantities. The average population sizes and diversities at the stationary regime are determined by

the baseline death rates: the larger the death rate, the smaller the population size and diversity. Also, both quantities have a positive correlation, see (**C**, **D**). As the populations evolve, the average payoffs $\langle A_{ij} \rangle$ increase, but the increment decreases, indicating a weak selection regime. The initial payoff is denoted by $A_0$. Particularly, the average increases logarithmically in the stationary regime. The left inset is the same data as the main panel on a linear-log scale and shows the logarithmic increase of the average payoffs. The right inset is the probability distribution function of all payoffs in the steady state for $\alpha = 5 \cdot 10^{-6}$. The distribution is smooth but slightly skewed to the left. ($\lambda_b = 0.9$, $\lambda_d = 0.4$, $\sigma^2 = 1$, and $\mu = 10^{-5}$).

## Appendix 3

### Link proportions

From the payoff matrix, we can determine the pairwise relationship between types, constructing a network. We represent the relationship between two types $i$ and $j$ with three different link types, dominance, bistability, and coexistence:

$$\begin{aligned}
i \leftarrow\leftarrow j \ \ \text{or} \ \ i \rightarrow\rightarrow j &: \text{Dominance of } i \text{ or } j, \\
i \leftarrow\rightarrow j &: \text{Bistability,} \\
i \rightarrow\leftarrow j &: \text{Coexistence.}
\end{aligned} \tag{A3.1}$$

As a basic element of a network, we investigate the frequencies of link types first.

As a reference, we consider a random payoff matrix model wherein all payoffs are randomly drawn from the standard normal distribution. In this case, the probability that one payoff is larger than another is 0.5 because all payoffs are independently sampled from the same distribution. Thus each condition in *Appendix 1—table 1* happens with the same probability, 0.25. Thus, the probability to observe a dominance link is 0.5 (since there are two directions), and the others are 0.25. We use these values as a reference to compare the link proportions obtained from population dynamics.

With population dynamics, the behavior of link frequencies differs for different $\alpha$. For small $\alpha$, the ensemble-averaged proportions are stable in time, while with large $\alpha$ values the average proportions fluctuate a lot, see *Appendix 3——figure 1A and B*. For large $\alpha$, the number of types rarely exceeds one. Hence, the link proportions fluctuate a lot, but the average values are well predicted from a random payoff matrix because the links appear by chance. On the other hand, for small $\alpha$, many links can be formed due to the presence of multiple types. In this case, the average link proportions are stable in the stationary regime, and coexistence links are favored compared to the random matrix case. We also plot the mean of ensemble-averaged link proportions in the stationary regimes $9500 \leq t \leq 10000$ for various $\alpha$ in *Appendix 3——figure 1A*. As we can see, the link proportions get closer to that of the random matrix as increasing the baseline death rates $\alpha$.

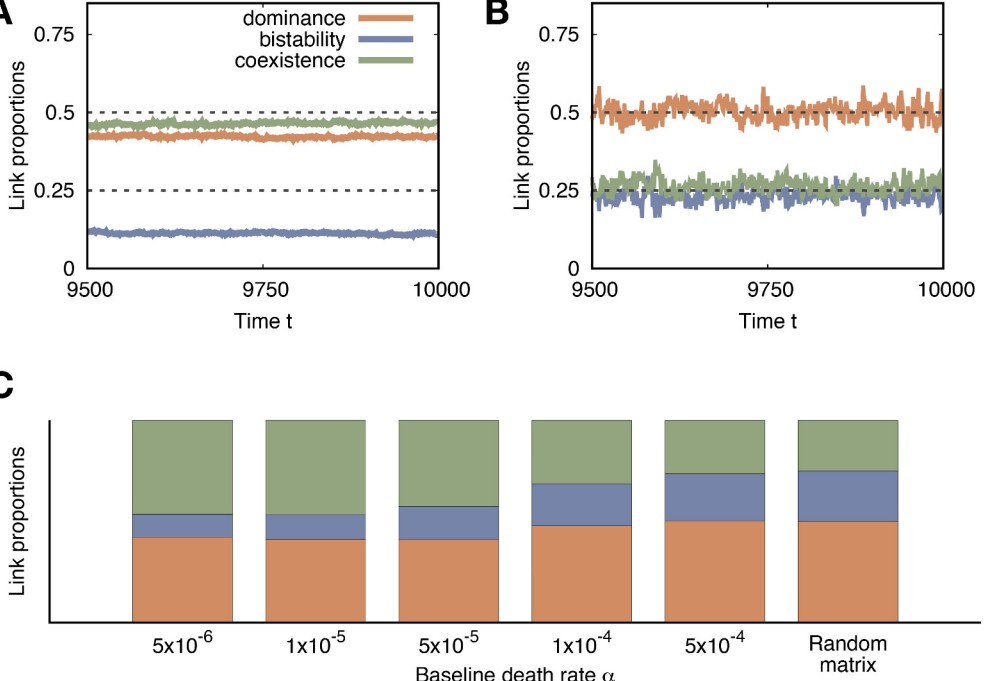

**Appendix 3—figure 1.** Average link proportions in the steady-state. Time series of the average link proportions in the stationary regime for small baseline death rates (**A**, $\alpha = 5 \cdot 10^{-6}$) and large baseline death rates (**B**, $\alpha = 5 \cdot 10^{-4}$). The average runs over all surviving samples. C The average link proportions in the steady-state indicate different values for different $\alpha$. We averaged those values in time to get the representative link proportions in the stationary regime, $9500 \leq t \leq 10000$. We find

that large $\alpha$ agrees with the values predicted by the random payoff matrix, because almost all links are formed by chance due to the low diversity. For small $\alpha$ where the population size and the diversity are larger, we find more coexistence links, which tend to be stable for a long time.

## Appendix 4

### Triplet proportions

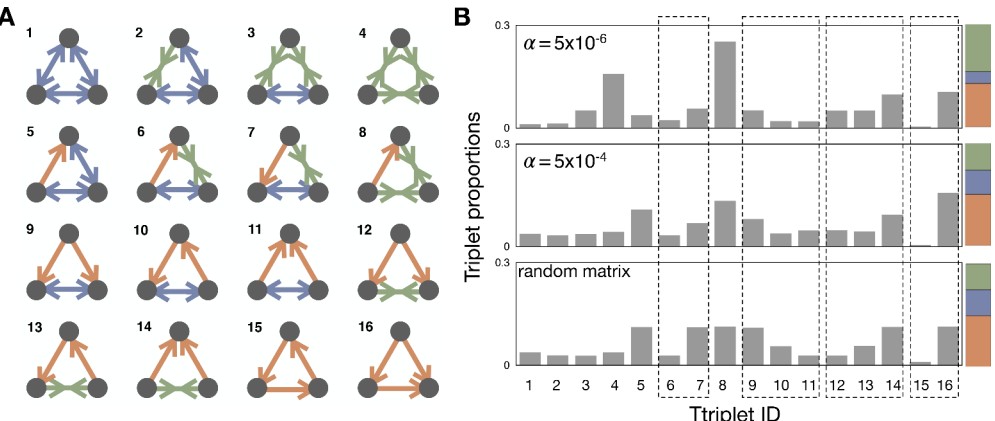

**Appendix 4—figure 1.** Triplet structures and their proportions. (**A**) With three different link types, dominance with directionality, bistability, and coexistence, we can find 16 different triplet structures. We draw possible triplets and label them with an integer index we will refer to as Triad ID. (**B**) The average proportions of triplets in the steady-state are shown in the population dynamics with $\alpha = 5 \cdot 10^{-6}$ and $\alpha = 5 \cdot 10^{-4}$ and the random matrix. Since the link proportions strongly affect the triplet proportions, the direct comparison of triplets that have different link composition is meaningless. Thus, we focus on a set of triplets, which have the same link composition but a different structure. There are four such sets (6-7; 9-11; 12-14; 15-16), and we marked using dashed boxes for those sets in the panel. The last box indicates the cyclic and non-cyclic dominance triplets. For comparison, the colored stacked bar shows the proportions of the different link types for the corresponding value of $\alpha$.

**Appendix 4—table 1.** Degeneracy of each triplet structure and their proportions in the random matrix.
Dashed boxes marks sets of triplets, which have the same link composition, but a different structure.

| Triad ID | Degeneracy | Proportions |
| --- | --- | --- |
| 1 | 1 | 4/108 |
| 2 | 3 | 3/108 |
| 3 | 3 | 3/108 |
| 4 | 1 | 4/108 |
| 5 | 6 | 12/108 |
| 6 | 6 | 3/108 |
| 7 | 6 | 12/108 |
| 8 | 6 | 12/108 |
| 9 | 3 | 12/108 |
| 10 | 6 | 6/108 |
| 11 | 3 | 3/108 |
| 12 | 3 | 3/108 |
| 13 | 6 | 6/108 |
| 14 | 3 | 12/108 |
| 15 | 2 | 1/108 |
| 16 | 6 | 12/108 |

Let us now consider triplets that are constructed by three links. With four kinds of link relationships, in total we can find 64 possible triplets. However, if we take into account symmetries, this reduces to 16 different triplet structure, see *Appendix 4—figure 1*. We call the number of triplets, which have the same triplet structure degeneracy. For example, with type indices $i$, $j$, and $k$, the cyclic dominance triplet structure can be found in two different triplets: $i \to\to j \to\to k \to\to i$ and $i \leftarrow\leftarrow j \leftarrow\leftarrow k \leftarrow\leftarrow i$. In contrast, there is no degeneracy for the triplet types marked by 1 and 4 in *Appendix 4—figure 1A* because changing the indices of types does not give any difference. Different triplet structures have different degeneracy because they have different mirror and rotation symmetries, summarized in *Appendix 4—figure 1*. Since our interest is the triplet structure, we investigate proportions of 16 triplet structures considering those degeneracies.

Again we use a random payoff matrix to attain the reference triplet proportions first. For three types $i$, $j$, and $k$, we can construct a 3 3 random matrix,

$$\mathbf{A} = \begin{pmatrix} A_{ii} & A_{ij} & A_{ik} \\ A_{ji} & A_{jj} & A_{ji} \\ A_{ki} & A_{kj} & A_{kk} \end{pmatrix}. \tag{A4.1}$$

All elements are independently drawn from the standard normal distribution $\mathcal{N}(x|0,1)$. Here, $\mathcal{N}(x|m,\sigma^2)$ indicates a Gaussian distribution with mean $m$ and variance $\sigma^2$. Any triplet is fully defined by three relationships, and each pairwise game determines one relationship. Mathematically, this corresponds to three $2 \times 2$ submatrices of $\mathbf{A}$. However, these matrices are not independent, as it would require $3 \times 2 \times 2 = 12$ independent parameters while the payoff matrix has only $3 \times 3 = 9$ entries.

To determine the type of triplet structure, we focus on one type of individual $j$. For this type $j$, we can characterize invasion possibilities against the two other types, giving four possible situations,

- $i \;\rightharpoonup j \rightharpoonup\; k$, when $A_{ij} < A_{jj} < A_{kj}$
- $i \;\leftharpoonup j \leftharpoonup\; k$, when $A_{ij} > A_{jj} > A_{kj}$
- $i \;\rightharpoonup j \leftharpoonup\; k$, when $A_{jj} > A_{ij}$ and $A_{jj} > A_{kj}$
- $i \;\leftharpoonup j \rightharpoonup\; k$, when $A_{jj} < A_{ij}$ and $A_{jj} < A_{kj}$

where $i \;\rightharpoonup j$ means that $j$ is stable with respect to invasion of $i$ from rare (either $j$ dominates $i$, $i \rightharpoonup\rightharpoonup j$ or they are bi-stable $i \leftharpoonup\rightharpoonup j$, hence the notation). Mathematically, this corresponds to three $3 \times 1$ columns of $\mathbf{A}$. Since all columns are independent, the probability of finding a certain triplet is the product of three probabilities to find a certain set of stubs for each type.

Thus we calculate the probability to find a certain set of subs for a type. The probability $2 \times 2$ that three random variables $a$, $b$, and $c$ satisfy the condition $a < b < c$ is given by convolution,

$$\begin{aligned} P(a) &= \int_{-\infty}^{\infty} \int_{-\infty}^{c} \int_{-\infty}^{b} \mathcal{N}(a|0,1)\mathcal{N}(b|0,1)\mathcal{N}(c|0,1) da\, db\, dc \\ &= \int_{-\infty}^{\infty} \int_{-\infty}^{c} \Phi(b)\mathcal{N}(b|0,1)\mathcal{N}(c|0,1) db\, dc \\ &= \frac{1}{2} \int_{-\infty}^{\infty} [\Phi(c)]^2 \mathcal{N}(c|0,1) dc \\ &= \frac{1}{6} \end{aligned} \tag{A4.2}$$

where $\Phi(x)$ is the probit function, $\Phi(x) = \int_{-\infty}^{x} \mathcal{N}(\theta|0,1) d\theta$. Hence, the probability that a type has in- and out-stubs is 1/6. On the other hand, the probability $P(c < a \,\text{and}\, c < b)$ is

$$\begin{aligned} P(c < a \,\text{and}\, c < b) &= \int_{-\infty}^{\infty} \int_{c}^{\infty} \int_{c}^{\infty} \mathcal{N}(a|0,1)\mathcal{N}(b|0,1)\mathcal{N}(c|0,1) da\, db\, dc \\ &= \int_{-\infty}^{\infty} [1 - \Phi(c)]^2 \mathcal{N}(c|0,1) dc \\ &= \frac{1}{3} \end{aligned} \tag{A4.3}$$

In the same way, $P(c > a \,\text{and}\, c > b)$ is 1/3. Hence, the probabilities to observe each set of stubs are

- $i \quad \rightarrow j \rightarrow \quad k$ appears with probability 1/6
- $i \quad \leftarrow j \leftarrow \quad k$ appears with probability 1/6
- $i \quad \rightarrow j \leftarrow \quad k$ appears with probability 1/3
- $i \quad \leftarrow j \rightarrow \quad k$ appears with probability 1/3

With those probabilities and the degeneracies, we can calculate the triplet proportions expected in a random payoff matrix. The results are summarized in *Appendix 4—table 1*, where we find the cyclic dominance (Triad ID 15) is the rarest one. At the same time, the non-cyclic dominance (Triad ID 16) is one of the most abundant triplets. From the proportions we can also obtain the fraction of cyclic dominance among cyclic and non-cyclic dominance triplets as $\chi = 1/13$, indicating 12 non-cyclic dominances can occur while only a single cyclic dominance appears.

Now we move to triplet proportions in population dynamics, see *Appendix 4—figure 1B*. Since the triplet proportions strongly depend on link proportions, the results for $\alpha = 5 \cdot 10^{-6}$ and $\alpha = 5 \cdot 10^{-4}$ are very different. For example, coexistence links are norm for $\alpha = 5 \cdot 10^{-6}$ and thus the triplet with Triad ID 4 is more abundant than in other cases. Thus, it is hard to directly compare proportions of triplets, which have different link composition. Instead of that, we compare the triplets, which have the same link composition. This comparison allows us to find which structure is more abundant, eliminating the effect of link proportions. There are four such sets of triplets, see *Appendix 4—figure 1B*. One of the sets consists of cyclic and non-cyclic dominance triplets, which are composed of three dominance links. If we look at the fraction $\chi$ of cyclic dominance, we can find that non-cyclic dominance is more suppressed in population dynamics than in the random matrix. For the other three sets of triplets, we can also find that the types which have bistability usually dominate another type while the types with coexistence are dominated by others. However, this tendency becomes weaker in population dynamics compared to the random payoff matrix analysis.

## Appendix 5

### Properties of payoff matrices for three types and a given genealogy

The emergence of a new mutant type $l$ induces a new row and a new column in the payoff matrix. If we denote the parental type of type $l$ as $r(l)$, the new payoffs are written by

$$
\begin{aligned}
A_{li} &= A_{r(l)i} + \xi(1),\\
A_{il} &= A_{ir(l)} + \xi(1),\\
A_{ll} &= A_{r(l)r(l)} + \xi(1),
\end{aligned}
\tag{A5.1}
$$

where $\xi(\sigma^2)$ are random values sampled from the Gaussian distribution with zero mean and variation $\sigma^2$, $\mathcal{N}(x|0,\sigma^2)$. Due to this inheritance, the genealogical structure which tells us who is whose parent type shapes the payoffs. In this section, we show how payoff elements are determined by a given genealogy.

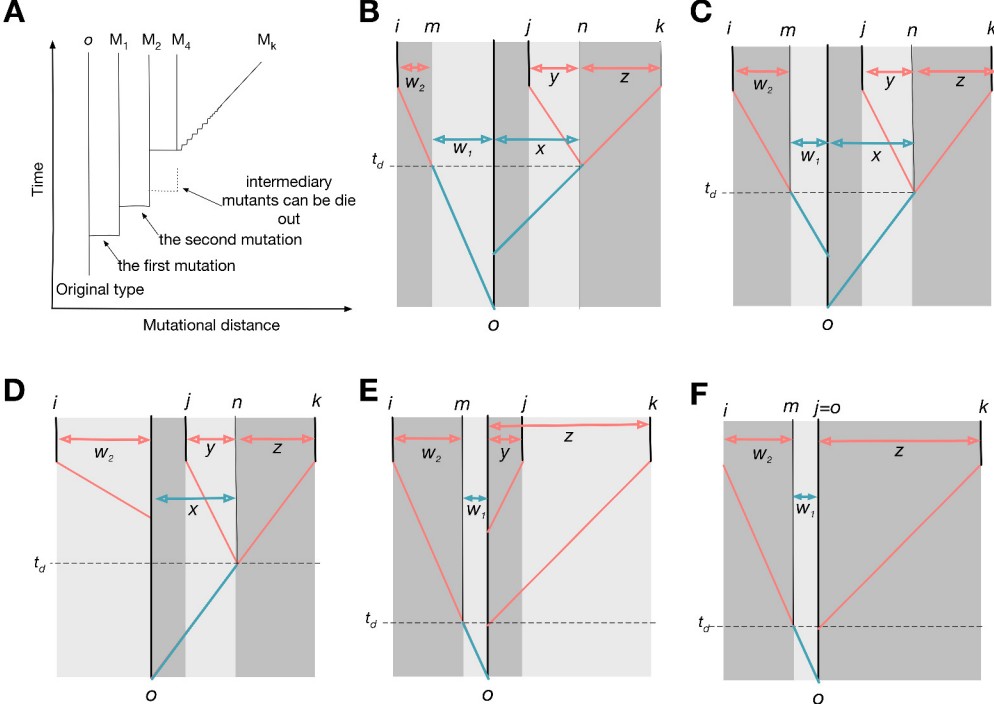

**Appendix 5—figure 1.** Possible genealogies for three types. (**A**) The scheme of genealogy. Vertical axis represents time, and horizontal axis represents mutational distance. Each solid vertical line corresponds to a surviving type. Horizontal lines indicate when mutation happens. Intermediary mutants can be omitted and a sequence of intermediary mutations is represented by a diagonal line. (**B-E**) Four possible genealogies of three types. Types $i$, $j$, and $k$ are focal types, and type $o$ is their last common ancestor. Type $n$ is the last common ancestor of $j$ and $k$, and type $m$ is the progenitor of type $i$ existed at the moment when type $j$ diverges from type $k$. The numbers of mutations between types are represented as $w_1, w_2, x, y$, and $z$: $o \to m$ is $w_1$; $m \to i$ is $w_2$; $o \to n$ is $x$; $n \to j$ is $y$; $n \to k$ is $z$.

Genealogies can be schematically depicted with time and mutational distance axes, see **Appendix 5—figure 1A**. For three focal types $i$, $j$, and $k$, a number of possible genealogies exists. We begin with a scenario shown in **Appendix 5—figure 1B**, where the lineage leading to the type $i$ diverges from the last common ancestor $o$ first and a different lineage leading to the two types $j$ and $k$ diverges from $o$ later. We index the last common ancestor of $k$ and $j$ in their lineage as $n$. Also, we give an index $m$ for the progenitor of type $i$ at the moment when lineages leading to types $j$ and $k$ diverged.

We begin with a payoff matrix of types $i$, $j$, and $k$ and trace it back to the moment of the last common ancestor type $o$ of all three types. All nine payoffs between types $i$, $j$, and $k$ are derived from the single payoff value $A_{oo}$.

For a pair of types $j$ and $k$, all four payoffs $A_{jj}$, $A_{jk}$, $A_{kj}$, and $A_{kk}$ can be traced back to the payoff $A_{nn}$. Let us imagine that $y$ and $z$ mutations happen leading the emergence of type $j$ and $k$ from the type $n$ respectively. If $y$ mutation events occur first before $z$ mutation events, we can write the value of $A_{jk}$ as

$$
\begin{aligned}
A_{jk} &= \\
&= A_{j,r(k)} + \xi(1) \\
&= A_{j,r^2(k)} + \xi(1) + \xi(1) \\
&\cdots \\
&= A_{jn} + \sum_{s=1}^{z} \xi(1) \\
&= A_{r(j),n} + \xi(1) + \sum_{s=1}^{z} \xi(1) \\
&\cdots \\
&= A_{nn} + \sum_{s=1}^{y+z} \xi(1).
\end{aligned}
\tag{A5.2}
$$

Since the random variable $\xi$ follows the normal distribution, we can simply write

$$
A_{jk} = A_{nn} + \xi(y+z).
\tag{A5.3}
$$

Thus, multiple mutation events can be written as a single mutation event with larger variance. For any other order of the mutation events, the final expression is the same even though all intermediate terms will be different.

In the same way, we can proceed with the other three payoffs. Consequently, all four payoffs can be written as

$$
\begin{aligned}
A_{jj} &= A_{nn} + \xi(y), \\
A_{jk} &= A_{nn} + \xi(y+z), \\
A_{kj} &= A_{nn} + \xi(y+z), \\
A_{kk} &= A_{nn} + \xi(z).
\end{aligned}
\tag{A5.4}
$$

Self-interactions, $A_{jj}$ and $A_{kk}$, do not change when the other type accumulates a mutation, so their mutational distances from $A_{nn}$ are smaller.

Next, we consider type $i$ which is diverged from the type $o$ but is a different lineage from $j$ and $k$. Then, the payoff $A_{ii}$ can be traced back to $A_{mm}$ as

$$
A_{ii} = A_{mm} + \xi(w_2),
\tag{A5.5}
$$

where $w_2$ is the number of mutations accumulated in the lineage of type $i$ from type $m$. The payoffs between $i$ and two other types can be traced back to payoffs between their progenitors $A_{mn}$ and $A_{nm}$

$$
\begin{aligned}
A_{ij} &= A_{mn} + \xi(w_2 + y), \\
A_{ik} &= A_{mn} + \xi(w_2 + z), \\
A_{ji} &= A_{nm} + \xi(w_2 + y), \\
A_{ki} &= A_{nm} + \xi(w_2 + z).
\end{aligned}
\tag{A5.6}
$$

Hence, all nine payoffs describing interactions between types $i$, $j$, and $k$ can be traced back to four payoffs, $A_{mm}$, $A_{mn}$, $A_{nm}$, and $A_{nn}$, characterizing interactions between $m$ and $n$. These four payoffs, in turn can be traced back to $A_{oo}$ in the same way,

$$
\begin{aligned}
A_{mm} &= A_{oo} + \xi(w_1), \\
A_{mn} &= A_{oo} + \xi(w_1 + x), \\
A_{nm} &= A_{oo} + \xi(w_1 + x), \\
A_{nn} &= A_{oo} + \xi(x),
\end{aligned}
\tag{A5.7}
$$

where $x$ is the number of mutations accumulated from the common ancestor $o$ to the type $n$, and $w_1$ is the number of mutations accumulated from the type $o$ to the type $m$. In summary, we show how payoffs can be calculated from their common ancestors payoffs, see **Appendix 5—figure 2**.

The same calculations apply to any other genealogy, for example ones presented in **Appendix 5—figure 1C-E**, if we label the types in a specific way. The labels $j$ and $k$ should indicate the pair of types, which diverged the last, that is, the last common ancestor of $j$ and $k$ is the most recent one among all three pairwise the last common ancestors. By exclusion, the rest one becomes type $i$. Type $n$ is the last common ancestor of $j$ and $k$ (for genealogy on **Appendix 5—figure 1E** and F, $n$ is the same as $o$). Finally, type $m$ is the progenitor of $i$ existed at the moment of divergence of $j$ and $k$ (for genealogy on **Appendix 5—figure 1D**, $m$ is the same as $o$).

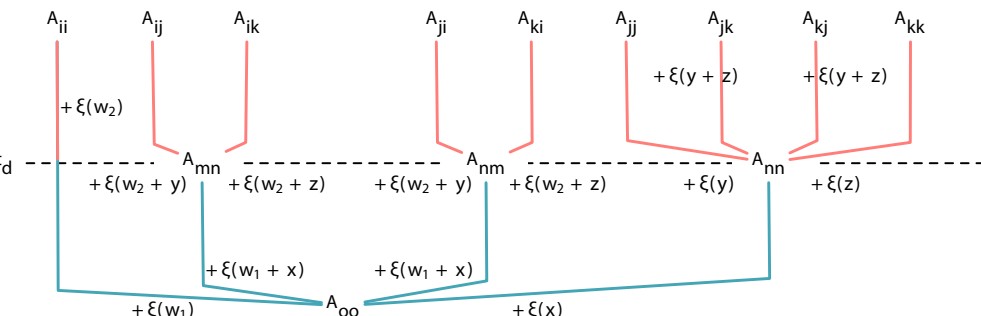

**Appendix 5—figure 2.** Expression of payoffs derived from its parental payoffs Each element of payoff matrix in a triplet can be traced back to $A_{oo}$. However, due to the genealogy structure, some payoffs have additional common ancestors later than $A_{oo}$. Note that for a genealogy at **Appendix 5—figure 1D**, $m = o$ and for **Appendix 5—figure 1E**, $n = o$. Adding up the random variable $\xi$ to the payoff in the lower part gives the payoff in the upper part.

## Appendix 6

### Minimizer and maximizer genealogies

In this section, we numerically identify those genealogies which produce the largest or smallest proportions of certain triplets. First of all, we note that the fractions of triplets do not depend on scales of the set of mutational distances $D_m = \{w_1, w_2, x, y, z\}$ without population dynamics; rescaling the distances results in the same triplet proportions. Therefore, without loss of generality, we can assume that all five mutational distances sum up to one and are thus located on the simplex $w_1 + w_2 + x + y + z = 1$.

We search for extreme genealogies by numerical optimization, implemented by a hill climbing algorithm. The optimization starts from a random set of $D_m$ and identifies the local optimum. Since our method can find only local optima, we use multiple initial values to find optimal set. After we identified the mutational distance sets which give local optima, we cluster them using principle component analysis (PCA). We denote the $X$-th class of genealogies that maximize and minimize the proportion of triplet *tri* by $D^{tri} - X$ and $D_{tri} - X$, respectively. For different scenarios, we find different numbers of distinct classes of genealogies.

### Maximization of the number of cyclic dominances

First, we ask which genealogies maximize the number of cyclic dominance triplets. Performing PCA analysis, we found five different classes of such genealogies, see *Appendix 6—table 1*. We denote cyclic dominance triplets as $T_{15}$ and non-cyclic dominance ones as $T_{16}$. Classes $D^{15} - 1$ and $D^{15} - 2$ are effectively the same genealogy, where all three types diverge from each other very early and then one of these types accumulates all the mutations (the most isolated type $i$ in $D^{15} - 1$ or one of the more recently diverged types $k$ in $D^{15} - 2$, see the end of Appendix 5 for the details of types labelling). Classes $D^{15} - 3$ and $D^{15} - 5$ also are also similar each other. All three types diverge from each other as early as possible and then two of them equally share the subsequent mutations ($j$ and $k$ in $D^{15} - 3$, or $i$ and $k$ in $D^{15} - 5$). In the class $D^{15} - 4$, all three types diverge from each other as early as possible. In this case, each of three types have the similar number of accumulated mutations (33%).

**Appendix 6—table 1.** Genealogies maximizing $T_{15}$.
Median values of each of five mutational distances $w_1, w_2, x, y, z$ are given. We use 12600 independent random initial values that each code one genealogy. We count how many of these genealogies belong to each class and sort classes by the average proportion of $T_{15}$.

| Class | Counts | Proportion of $T_{15}$ | $w_1$ | $w_2$ | $X$ | $Y$ | $Z$ |
|---|---|---|---|---|---|---|---|
| $D^{15} - 1$ | 5044 | 0.020 | 0.0020 | 0.98 | 0 | 0 | 0.010 |
| $D^{15} - 2$ | 2012 | 0.019 | 0 | 0 | 0 | 0 | 0.98 |
| $D^{15} - 3$ | 683 | 0.016 | 0 | 0.0021 | 0.0017 | 0.37 | 0.59 |
| $D^{15} - 4$ | 2701 | 0.014 | 0.0058 | 0.32 | 0 | 0.23 | 0.41 |
| $D^{15} - 5$ | 2160 | 0.016 | 0.0046 | 0.52 | 0 | 0 | 0.44 |

### Minimization of the number of non-cyclic dominances

Next, we ask which genealogies instead minimize the number of non-cyclic dominance triplets, with the expectation that they will be similar to the ones above. Performing PCA analysis, we again find five different classes, see *Appendix 6—table 2*. The obtained genealogies minimizing $T_{15}$ have the same characteristics of mutational distances with the genealogies maximizing $T_{16}$, see section F. The genealogy class $D_{16} - 1$ is the same with $D^{15} - 1$. The class $D_{16} - 2$ is the same as $D^{15} - 2$, and $D_{16} - 3, 4, 5$ are the same as $D^{15} - 4$. Classes $D_{16} - 3$, $D_{16} - 4$, and $D_{16} - 5$ differ only in their values of $w_1$ and $x$. Thus

these can be considered as a fine structure of a single class. Classes equivalent to $D^{15}-3$ and $D^{15}-5$ are not found in $D_{16}$.

**Appendix 6—table 2.** Genealogies minimizing $T_{16}$.
Median values of $D_m$ are given. We use 20000 independent simulations.

| Class | Counts | Proportion of $T_{16}$ | $w_1$ | $w_2$ | X | Y | Z |
|---|---|---|---|---|---|---|---|
| $D_{16}-1$ | 1576 | 0.11 | 0.0045 | 0.97 | 0 | 0 | 0.012 |
| $D_{16}-2$ | 697 | 0.11 | 0.0012 | 0.0021 | 0 | 0.011 | 0.96 |
| $D_{16}-3$ | 5883 | 0.11 | 0 | 0.29 | 0.019 | 0.22 | 0.45 |
| $D_{16}-4$ | 598 | 0.11 | 0.019 | 0.29 | 0.014 | 0.23 | 0.44 |
| $D_{16}-5$ | 11246 | 0.11 | 0.034 | 0.31 | 0 | 0.21 | 0.43 |

## Minimization of the number of cyclic dominances

Alternatively, we can also ask for which genealogies it is hardest to obtain cyclic dominances. Performing PCA analysis, we find three different classes, see *Appendix 6—table 3*. In the class $D_{15}-1$, most of mutations are accumulated in the lineage of the type $i$ before types $j$ and $k$ are diverged. The divergence of types $j$ and $k$ tends to be the last mutational event in this genealogy. In the class $D_{15}-2$, most mutations are accumulated in the lineage of the common progenitor of types $j$ and $k$ before their divergence. The divergence of types $j$ and $k$ tends to be the last mutational event in this genealogy. In the class $D_{15}-3$, mutations are equally shared between the lineage of type $i$ and the common progenitors of types $j$ and $k$ before their divergence. Similar to above two cases, the divergence of types $j$ and $k$ tends to be the last mutational event in this genealogy.

**Appendix 6—table 3.** Genealogies minimizing $T_{15}$.
Median values of $D_m$ are given. We use 20000 independent simulations.

| Class | Counts | Fraction of $T_{15}$ | $w_1$ | $w_2$ | X | Y | Z |
|---|---|---|---|---|---|---|---|
| $D_{15}-1$ | 4995 | $<10^{-4}$ | 0.97 | 0 | 0.018 | 0 | 0.0051 |
| $D_{15}-2$ | 5011 | $<10^{-4}$ | 0.014 | 0 | 0.97 | 0 | 0.0057 |
| $D_{15}-3$ | 9994 | 0.00012 | 0.48 | 0 | 0.49 | 0.0 | 0.014 |

## Maximization of the number of non-cyclic dominances

Now we ask for which genealogies it is easiest to obtain non-cyclic dominances. Performing PCA analysis, we find again three different classes of geneaolgies, see *Appendix 6—table 4*. Again, the obtained three classes are the same as minimizing cyclic dominances $T_{15}$: The class $D^{16}-1$ is the same as $D_{15}-1$, and $D^{16}-2$ is the same as $D_{15}-2$, and $D^{16}-3$ is the same as $D_{15}-3$.

**Appendix 6—table 4.** Genealogies maximizing $T_{16}$.
Median values of $D_m$ are given.

| Class | Counts | Proportion of $T_{16}$ | $w_1$ | $w_2$ | X | Y | Z |
|---|---|---|---|---|---|---|---|
| $D^{16}$-1 | 4772 | 0.24 | 0.97 | 0 | 0.025 | 0 | 0.0019 |
| $D^{16}$-2 | 11017 | 0.24 | 0.019 | 0 | 0.98 | 0 | 0.0023 |
| $D^{16}$-3 | 4211 | 0.24 | 0.45 | 0 | 0.53 | 0.0 | 0.0037 |

## Summary for optimization

From all optimization results we find two extreme genealogies for maximizing and minimizing the fraction $\chi$ of cyclic dominances. In the first class, most of mutations occur after all three lineages separate and these mutations are accumulated in a single lineage, see *Appendix 6—table 5*. We call

them maximizer genealogies. These genealogies promote cyclic dominances and suppress non-cyclic dominances. In the second class, the most of mutations occur before types $j$ and $k$ diverge. These genealogies suppress cyclic dominances, while promoting non-cyclic dominances. We call them maximizer genealogies.

**Appendix 6—table 5.** Minimizer and Maximizer genealogies.
The fraction $\chi$ of cyclic dominance is also calculated for each case.

| Matrix generation | $T_{15}$ | $T_{16}$ | |
|---|---|---|---|
| Random matrix | 0.0093 | 0.11 | 0.077 |
| Maximizer | 0.02 | 0.11 | 0.154 |
| Minimizer | $< 10^{-4}$ | 0.24 | 0.000 |

## Analytics for maximizer genealogies

We calculate the proportions of cyclic and non-cyclic dominance triplets at maximizer genealogies. In that case, the cyclic dominance triplets occur

- $i \rightarrow j \rightarrow k \rightarrow i$ with probability $1/96$
- $i \rightarrow k \rightarrow j \rightarrow i$ with probability $1/96$

Altogether, this results in $p(T_{15}) = 1/48 \approx 0.0208$. The non-cyclic dominance triplet $T_{16}$ has degeneracy six, and they occur

- $j \rightarrow k \rightarrow i$ with probability $1/96$
- $i \rightarrow k \rightarrow j$ with probability $1/96$
- $k \rightarrow j \rightarrow i$ with probability $1/48$
- $k \rightarrow i \rightarrow j$ with probability $1/48$
- $j \rightarrow i \rightarrow k$ with probability $1/48$
- $i \rightarrow j \rightarrow k$ with probability $1/48$

Altogether, this results in $p(T_{16}) = 5/48 \approx 0.104$, yielding $\chi = 1/6 \approx 0.167$. The results agree well with numerical results in *Appendix 6—table 5*.

## Minimal model

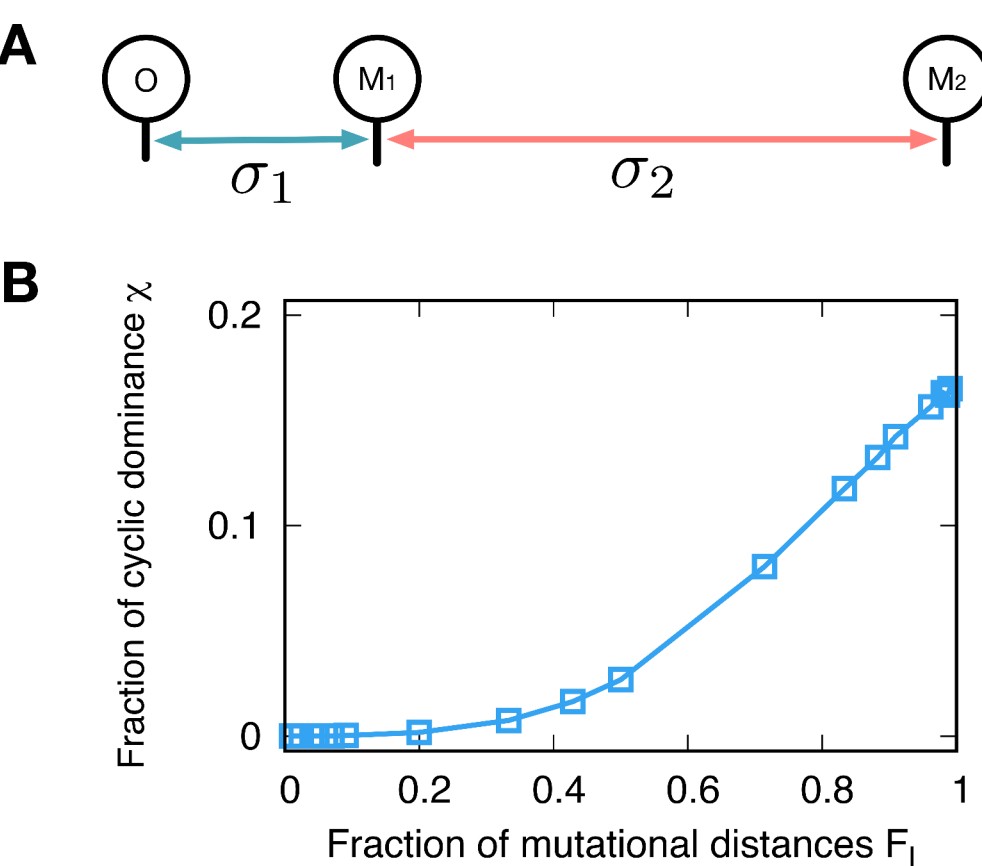

**Appendix 6—figure 1.** Minimal model. (**A**) We can also shape the payoff correlation by controlling the sampling distribution. We use different variances of Gaussian noise for different mutation events, controlling the mutational distances. Mutational distances are controlled by not only how many steps proceed but also how big the jump is related variance of noise. Here, we assume a very simple genealogy type with different noise variances; The first mutant type $M_1$ originates from the type $o$ with variance $\sigma_1^2$ while the second mutant $M_2$ mutated from $M_1$ has variance $\sigma_2^2$ for new payoffs. (**B**) Since only the relative mutational distances matter, we defined $F_l = \frac{\sigma_2}{\sigma_1 + \sigma_2}$. As we vary the fraction $F_l$, we calculate the fraction $\chi$ for three types $o$, $M_1$, and $M_2$. Small $F_l$ values give the same payoff structure of the minimizer while the maximizer is reproduced for large $F_l$. The chance to observe the cyclic dominance triplets increases with $F_l$, because types $M_1$ and $M_2$ become uncorrelated.

Besides genealogies, there is also another way to shape the payoff correlation. Mutational distances are controlled by not only how many steps proceed but also how big the jump is related variance of noise. In a minimal model, we can directly control the closeness of new-born type by using a different variance $\sigma^2$ of Gaussian noise. We generate the first mutant type $M_1$ from the original type $o$ with the normal distribution, and we use variance $\sigma_1^2$ when we generate the type $M_1$. Then, we assume that $M_1$ is the parental type of type $M_2$, and the variance $\sigma_2^2$ is used in this case. This procedure is schematically drawn in **Appendix 6—figure 1A**. Note that only the relative mutational distances are important, and thus we introduce the fraction $F_l$ that measures the distances before and after the divergence of the third type. When $F_l$ is small, the close type to the resident type arises which is corresponding to the minimizer genealogy. On the other hand, if we use large $F_l$, it will reproduce the maximizer by introducing uncorrelated types in the population. The minimum and maximum average fractions $\chi$ are obtained by changing $F_l$, yielding almost zero and 0.165 respectively, see **Appendix 6—figure 1B**. These results agree well with the results of the genealogy approach.

## Appendix 7

### Correlation between diversity and triplet fraction

We investigated the correlation between diversity and the probability of having cyclic and non-cyclic dominance triplets in the system at the steady-state. We measured the Pearson correlation coefficient $r_{xy}$ between two variables $x$ and $y$ defined as

$$r_{xy} = \frac{\sum_i (x_i - \bar{x})(y_i - \bar{y})}{\sqrt{\sum_i (x - \bar{x})^2} \sqrt{\sum_i (y - \bar{y})^2}}, \tag{A7.1}$$

where $\bar{x}$ and $\bar{y}$ indicate the averages of the variables $x$ and $y$. The index $i$ runs over all samples (in our case, 1944 surviving realizations are used). The correlation coefficient is bound from -1 to 1, representing fully anti-correlated to fully-correlated variables.

We used three different diversity indices: (1) richness $n$ as the number of different types, (2) Shannon diversity index $H = -\sum_{i=1}^{n} f_i \ln f_i$, and (3) Simpson diversity index $S = \left(\sum_{i=1}^{n} f_i^2\right)^{-1}$. Note that a population frequency of type $i$ is denoted by $f_i$. Correlations were measured between each diversity index and the probabilities to having cyclic or non-cyclic dominance triplets, $P_{cyc}$ and $P_{ncyc}$, see *Appendix 7—figure 1*. All correlation coefficients are summarized in *Appendix 7—table 1*. The results show weak anti-correlation between diversity and cyclic dominance and stronger anti-correlation between diversity and non-cyclic dominance. This tendency does not change even when population frequencies are considered. Finally we measured the correlation coefficient between the fraction of cyclic dominance $\chi$ and diversity, and surprisingly the coefficient showed almost zero. This implies that there is no significant correlation between cyclic dominance and diversity.

**Appendix 7—table 1.** Pearson correlation between diversity and fraction of cyclic and non-cyclic triplets.

In total 1944 realizations are used for the calculations and the average is across 500 time steps. We find a weak anti-correlation between the fraction of cyclic dominance and diversity while the fraction of non-cyclic dominance has stronger anti-correlation with diversity. Despite these differences, almost no correlation between the fraction and diversity is found.

|  | Richness index $N$ | Shannon index $H$ | Simpson index $S$ |
|---|---|---|---|
| $P_{cyc}$ | -0.05 | -0.12 | -0.11 |
| $P_{ncyc}$ | -0.15 | -0.35 | -0.34 |
| $P_{cyc}f_{cyc}$ | -0.04 | -0.08 | -0.07 |
| $P_{ncyc}f_{ncyc}$ | -0.12 | -0.35 | -0.34 |
|  | 0.01 | -0.01 | -0.004 |

Anti-correlation between the fraction of non-cyclic dominance $P_{ncyc}$ and diversity could be understood from the population frequencies of non-cyclic dominance $f_{ncyc}$. In the steady-state, non-cyclic dominance usually occupies the majority of the population, taking around 0.83 of the population on average, see *Appendix 7—figure 2AB*. It seems that the non-cyclic dominance emerges from the majority, and thus their fraction could be high when the winner takes over almost the whole population, leading to low diversity. On the other hand, cyclic dominance usually takes only a minority, showing a peak near $f_{cyc} \approx 0$ in *Appendix 7—figure 2C*.

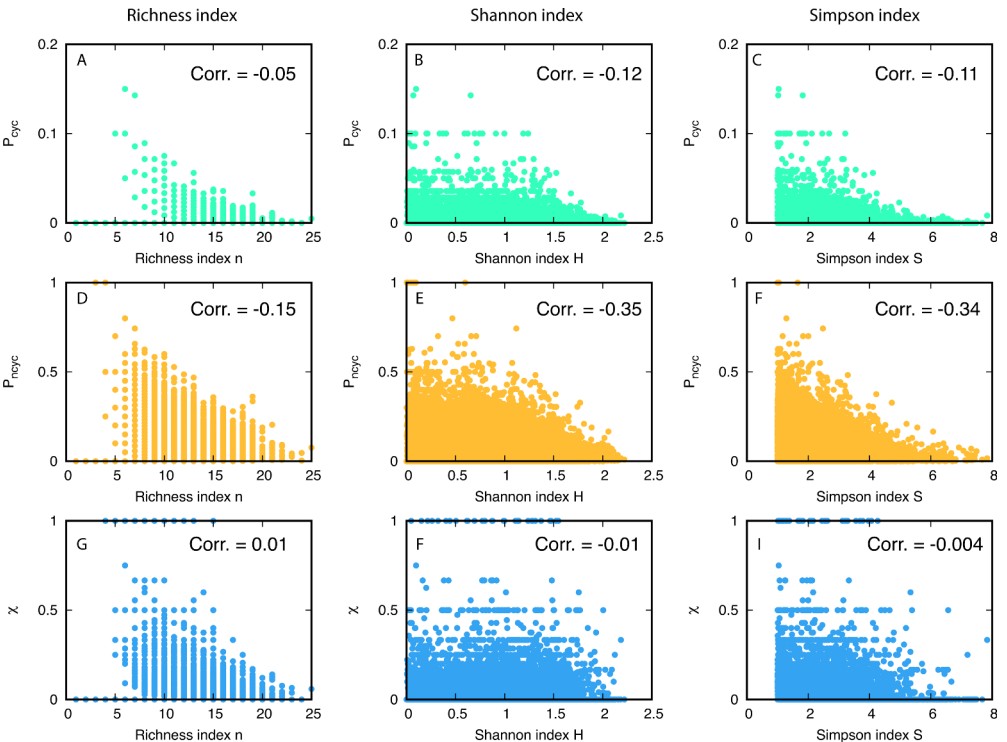

**Appendix 7—figure 1.** Scatter plots of diversity versus fractions of triplets. In the first and the second rows, we plot $P_{cyc}$ and $P_{ncyc}$ for various diversities (in each column) in the steady-state. The last row shows the scatter plot for $\chi = \frac{P_{cyc}}{P_{cyc}+P_{ncyc}}$. For 1944 realizations, we measured the Pearson correlation coefficients and noted the correlation measured in each panel.

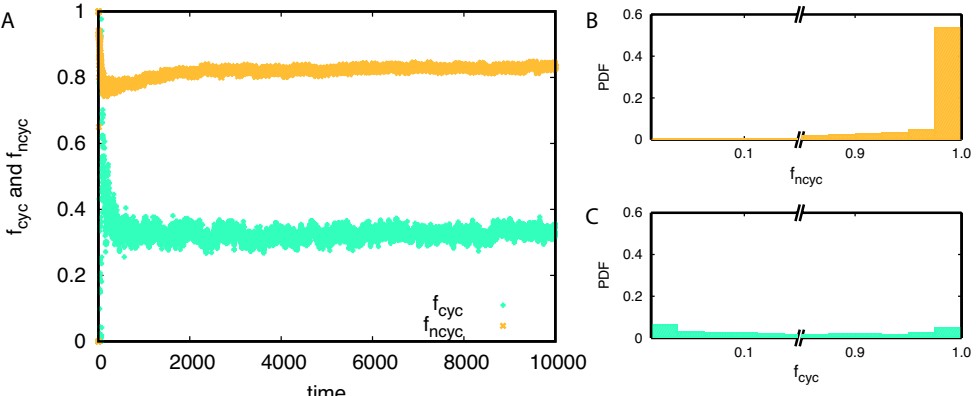

**Appendix 7—figure 2.** Average population frequencies belonging to cyclic and non-cyclic triplets $f_{cyc}$ and $f_{ncyc}$ over time. (**A**) On average around 83% of population belongs to non-cyclic dominance triplets while only 33% population belongs to cyclic dominance triplets. (**B-C**) The probability distribution functions of $f_{ncyc}$ and $f_{cyc}$ are shown for each triplet. Non-cyclic dominance mainly emerges from majorities while cyclic dominance usually takes small population frequencies. Only 30% of the time, cyclic dominance can take over more than half of the population.

