## [Decision Letter]

**Acceptance summary:**

The reviewers and I found your article to be an insightful perspective regarding why non-transitive (e.g. rock-paper-scissors) interactions might be rare in natural communities, despite the potential stabilizing effects that non-transitive interactions could have in sustaining diversity. In particular, your approach of analyzing the evolutionary origin of different interactions provides fascinating insight to an important problem.

**Decision letter after peer review:**

Thank you for submitting your article "Why is cyclic dominance so rare?" for consideration by *eLife*. Your article has been reviewed by two peer reviewers, including Jeff Gore as the Reviewing Editor and Reviewer #1, and the evaluation has been overseen by Patricia Wittkopp as the Senior Editor. The following individual involved in review of your submission has agreed to reveal their identity: Matthieu Barbier (Reviewer #2).

The reviewers have discussed the reviews with one another and the Reviewing Editor has drafted this decision to help you prepare a revised submission.

We would like to draw your attention to changes in our revision policy that we have made in response to COVID-19 (https://elifesciences.org/articles/57162). Specifically, we are asking editors to accept without delay manuscripts, like yours, that they judge can stand as *eLife* papers without additional data, even if they feel that they would make the manuscript stronger. Thus the revisions requested below only address clarity and presentation, but we encourage you to also consider the substantive points raise below.

Summary:

Cyclic dominance has been proposed as a possible stabilizing mechanism for diversity in communities. However, empirical evidence has suggested that cyclic dominance appears to be rare. Why might this be? The authors' explanation for this observation, based on a dynamical eco-evo model and various theoretical arguments, can be summed up as follows:

– When drawing a network of qualitative dominance relations, it may seem that cyclic dominance could happen often (1/4th of possible triplets if relations are drawn at random).

– But when the interactions arise from an underlying matrix of payoffs, the conditions on payoffs that create cyclical dominance are actually quite constraining, happening at random only in 1/13th of triplets of dominance links (which themselves are only 1/10th of triplets, since dominance is not the only outcome: one can also observe coexistence or bistability).

– And when the payoffs themselves arise from a continuous evolutionary process (where new types are introduced with small changes in payoffs from their ancestors), the probability is even lower. The authors' simulations show that it occurs only in 1/30th of dominance triplets.

The authors provide arguments for why this probability is even lower: it is due to correlations between parents and offsprings. They show in their simulation results that cyclical dominance is associated with more difference between types (measured as an overall difference in their payoffs) than non-cyclical dominance. They also run a simple genealogy process for a 3-type subcommunity (where, instead of each mutation leading to a new type that competes with all the existing types, mutations are now accumulated along genealogical branches as if, for every mutation, the parent type is automatically replaced) to show that cyclical dominance can be favored when many mutations occur after the branching out of the 3 types, thus decorrelating their properties.

Essential revisions:

1) The authors argue the importance of studying cyclic dominance by noting its potential to increase diversity. This paper would, therefore, be strengthened by including an analysis of whether diversity increased in simulations in which cyclic dominance emerged (or at time points during which it was present) as compared to simulations or time points in/at which there was no cyclic dominance. If there is no correlation between the emergence of cyclic dominance and increases in diversity, the authors should address this in the Discussion section. Relatedly, the authors claim in the Discussion "our results indicate cyclic dominance can support diversity over long time scales". We do not believe the first half of this claim (that cyclic dominance can support diversity) is currently supported by their results (it is rather from other literature).

2) For readers to make sense of the triplet analysis it is necessary to know the link frequencies (e.g. between dominance, coexistence, and bistability). This can be brief (no need for a full supplementary figure in main text).

3) The authors argue that it is difficult to evolve cyclic dominance due to mutants being similar to parents. The paper would therefore be strengthened by including an analysis of how the mutation size sigma influences the frequency of cyclic dominance. This is shown for the toy model in Appendix 6, but not for the main model. Many readers would likely appreciate the results of varying sigma within the main model.

4) On a related note, the authors could directly measure and report on the time passed and/or number of mutations accrued since species in cyclic dominance triplets diverged from their last common ancestor and make appropriate comparisons. The current section "Genealogical structure can promote and can suppress cyclic dominance" could be moved to the Appendix if a direct tracing of the genealogy of triplets is added to the paper. (We found the genealogy section to be a bit difficult to parse).

5) The authors could include a brief analysis of how often the species in cyclic dominance triplets gain large population fractions after emerging. This question is relevant because if the three species in a triplet never reach large population fractions then their effects on each other would presumably be small compared to the effects from other members of the population (which would not affect any of the conclusions of this paper but may affect some readers' interpretations of those conclusions).

6) We believe that an outline of the whole argument (a bit like the summary that we give above) should be introduced early in the article, so that the structure is clear: currently, the text reads like a journey where we discover things on the way, but it is not always clear at first why we are doing one thing or another. In particular, something that may not be an issue for theoretical readers, but would be for a more general audience, is that it is not obvious at first which aspects of the model are going to be important or not for the argument, and therefore, whether the results are specific to the model assumptions. In summarizing the results above, we tried to present them in a way that clarifies that a large part of the argument could be made without invoking any specific simulation model. Readers may worry about the interpretation or lack of realism of some assumptions of the dynamics, when they do not really matter for the argument. This is an issue at different points in the paper: for the general eco-evo model, but also for the genealogy "mini-model", where a reader could find it concerning that the process is now different from the main simulation (the text makes it sound as if a type can accumulate mutations without branching out). It would be helpful for the genealogy section to introduce a quick outline of the argument it will make. We also suggest showing the Appendix 5—figure 1A in the main text, maybe as a panel in Figure 5. It would help understand what this mini-model assumes compared to the main simulation model (that all branches except for 3 have disappeared or can be ignored, thus explaining what it means to "accumulate mutations along a branch").

7) Regarding the main simulation model, there is in fact a concern that does not impact the results. A common issue in eco-evo modela is that the evolving traits will keep changing in the same direction, which is generally avoided by imposing some trade-off.

As the authors note, the average payoff here tends to increase indefinitely, which means that all competition coefficients d_ij_ eventually converge to just α. In ecological terms, we would call that going toward neutral dynamics (where all individuals have the same competition strength, both within and between types, see S.P. Hubbell's 2001 book). This issue is not deadly because, as differences of d_ij_ between types become smaller, all that really happens is that the time it takes for a type to exclude another diverges. But the qualitative nature of the relationships is not changed: there is still dominance or coexistence or bistability, even if by a very small margin. This is simply impractical because one must wait longer and longer for extinctions to actually happen (this is actually theorized in ecology: types could coexist by becoming so similar than one winning over the other takes forever, see Scheffer and van Nes, 2006). So it seems to me that the simulation would have been easier if the payoffs were relative, e.g. constantly setting the average at zero, so that interactions do not converge to neutrality.

In the same spirit:

"For small α values (rich environments) in particular, the population size N at the steady state becomes large, containing many different types " the value of α should change nothing except the biomass scale (population size N ~ M lambda/alpha – incidentally, why introduce parameter M at all? it is never explained). The fact that more types exist for lower α seems like an artefact, perhaps because it takes longer for interactions to tend toward neutrality.

8) Perhaps one could directly show how adding correlations in the matrix diminishes the occurrence of cyclical dominance? This would directly show that nothing else is needed (i.e. that the eco-evo process does nothing more than add these correlations).

9) It is worth noting that there is a large ecological literature about extensions of cyclic dominance to more than triplets, called intransitive competition. Some of that literature has claimed (although we are not necessarily convinced) that intransitive competition is actually common and important. We would suggest reading these articles and figuring out why their claims would be so different – and if that difference is important, then discussing it in the article.

See for instance as starting points: Soliveres et al., 2015; Gallien et al., 2017.

---

## [Author Response]

Essential revisions:1) The authors argue the importance of studying cyclic dominance by noting its potential to increase diversity. This paper would, therefore, be strengthened by including an analysis of whether diversity increased in simulations in which cyclic dominance emerged (or at time points during which it was present) as compared to simulations or time points in/at which there was no cyclic dominance. If there is no correlation between the emergence of cyclic dominance and increases in diversity, the authors should address this in the Discussion section. Relatedly, the authors claim in the Discussion "our results indicate cyclic dominance can support diversity over long time scales". We do not believe the first half of this claim (that cyclic dominance can support diversity) is currently supported by their results (it is rather from other literature).

Thank you for this comment. Indeed, we have started out from the concept of diversity as it has often been stated as a crucial consequence of cyclic dominance. However, we have only shown the rareness of cyclic dominance and discussed mechanisms leading to it. As you pointed out, we did not explicitly show that cyclic dominance increases diversity. We briefly mentioned the lifetime of both cyclic and non-cyclic dominance triplets but we did not explicitly analyze correlations between triplets with diversity. We have now explored these correlations and added the detailed results in Appendix 7 and argued them in the Discussion. In a nutshell, we found that the fraction of cyclic dominance \chi has almost no correlation with diversity. This means – in contrast to general believe – that diversity is not mainly supported by the emergence of cyclic dominance in our model. However, the long time to equilibration with the removal of all non-competitive types and a short time to the emergence of new types are main drivers of diversity.

2) For readers to make sense of the triplet analysis it is necessary to know the link frequencies (e.g. between dominance, coexistence, and bistability). This can be brief (no need for a full supplementary figure in main text).

Agreed, we have added an explanation for link frequencies in the main text before the Results section. In addition, we have distinguished the terms “frequency” and “fraction”

preventing confusion. Frequency is only used when the population frequency is considered. In all other cases, we have used “proportion” or “fraction”.

3) The authors argue that it is difficult to evolve cyclic dominance due to mutants being similar to parents. The paper would therefore be strengthened by including an analysis of how the mutation size sigma influences the frequency of cyclic dominance. This is shown for the toy model in Appendix 6, but not for the main model. Many readers would likely appreciate the results of varying sigma within the main model.

In our toy model, the mutational distances, between o and M_1_ and between M_1_ and M_2_, are controlled by different variances while they are controlled by the number of mutations in the main model. In both cases, the absolute scale does not affect the results but the ratio of mutational distances is important. Hence, changing sigma in the main model affects the overall mutational distance, but the ratio remains unchanged, keeping the fraction of cyclic is the same. We have clarified this by discussing two explicit sigma values in the toy model.

4) On a related note, the authors could directly measure and report on the time passed and/or number of mutations accrued since species in cyclic dominance triplets diverged from their last common ancestor and make appropriate comparisons. The current section "Genealogical structure can promote and can suppress cyclic dominance" could be moved to the Appendix if a direct tracing of the genealogy of triplets is added to the paper. (We found the genealogy section to be a bit difficult to parse).

Thank you. Following this suggestion, we have moved the previous genealogy analysis into the Appendix and instead mentioned the direct measure of genealogical properties found in the simulations. The key variable to find cyclic or non-cyclic triplets is the fraction of accumulated mutational distances before and after the payoff correlation appears during diverging process, which agrees well with the toy model result. We have included a new Figure 5 to explain it and have Introduced F_l_ defined by the fraction of mutational distances. Addressing major comment 6, in the new figure we first explained the genealogy. Then we show an example of genealogy for three types and the relationship between the fraction \chi and F__l_. As many mutations are accumulated after each type evolves independently from others, the chance to emerge the cyclic dominance systematically increases.

5) The authors could include a brief analysis of how often the species in cyclic dominance triplets gain large population fractions after emerging. This question is relevant because if the three species in a triplet never reach large population fractions then their effects on each other would presumably be small compared to the effects from other members of the population (which would not affect any of the conclusions of this paper but may affect some readers' interpretations of those conclusions).

Thank you for this very interesting suggestion. We have measured the population frequencies which belong to cyclic and non-cyclic dominance, f_cyc_ and f_ncyc_. On average, cyclic dominance takes around 33% of the population, while non-cyclic dominance takes around 83%. To check how often cyclic dominance can take more than a half population, we have checked the probability distribution function (PDF) of f_cyc_ and f_ncyc_ at the steady-state. Cyclic dominance usually takes very low population frequencies or rarely take the majority, showing a bimodal distribution with two peaks around 0 and 1. The peak near 0 is higher than the other, implying the cyclic dominance usually emerges in small groups. Also, we found that a chance that cyclic dominance takes more than a half of the population is around 30%. On the other hand, the non-cyclic dominance usually takes the majority and 85% chance to take over the half population. These results have addressed in a new Appendix 7 and have shortly mentioned it in the Discussion.

6) We believe that an outline of the whole argument (a bit like the summary that we give above) should be introduced early in the article, so that the structure is clear: currently, the text reads like a journey where we discover things on the way, but it is not always clear at first why we are doing one thing or another. In particular, something that may not be an issue for theoretical readers, but would be for a more general audience, is that it is not obvious at first which aspects of the model are going to be important or not for the argument, and therefore, whether the results are specific to the model assumptions. In summarizing the results above, we tried to present them in a way that clarifies that a large part of the argument could be made without invoking any specific simulation model. Readers may worry about the interpretation or lack of realism of some assumptions of the dynamics, when they do not really matter for the argument. This is an issue at different points in the paper: for the general eco-evo model, but also for the genealogy "mini-model", where a reader could find it concerning that the process is now different from the main simulation (the text makes it sound as if a type can accumulate mutations without branching out). It would be helpful for the genealogy section to introduce a quick outline of the argument it will make. We also suggest showing the Appendix 5—figure 1A in the main text, maybe as a panel in Figure 5. It would help understand what this mini-model assumes compared to the main simulation model (that all branches except for 3 have disappeared or can be ignored, thus explaining what it means to "accumulate mutations along a branch").

Thank you for this very helpful suggestion. Based on the summary provided by the reviewers, we have now added an outline after making a point of the rareness of cyclic dominance. Also, taking a suggestion, we have moved Appendix 5—figure 1A to a new Figure 5A in the main text.

7) Regarding the main simulation model, there is in fact a concern that does not impact the results. A common issue in eco-evo model is that the evolving traits will keep changing in the same direction, which is generally avoided by imposing some trade-off. As the authors note, the average payoff here tends to increase indefinitely, which means that all competition coefficients d_ij_ eventually converge to just α. In ecological terms, we would call that going toward neutral dynamics (where all individuals have the same competition strength, both within and between types, see S.P. Hubbell's 2001 book). This issue is not deadly because, as differences of d_ij_ between types become smaller, all that really happens is that the time it takes for a type to exclude another diverges. But the qualitative nature of the relationships is not changed: there is still dominance or coexistence or bistability, even if by a very small margin. This is simply impractical because one must wait longer and longer for extinctions to actually happen (this is actually theorized in ecology: types could coexist by becoming so similar than one winning over the other takes forever, see Scheffer and van Nes, 2006. So it seems to me that the simulation would have been easier if the payoffs were relative, e.g. constantly setting the average at zero, so that interactions do not converge to neutrality.

We agree with your point. In our model, species or types can coexist because the system is close to the neutral regime. However, our finding (rareness of cyclic dominance) is robust even with rescaling of payoffs. This is because the main driver of the diversity in that case is also not the formation of cyclic dominance. Cyclic dominance cannot be stably sustained, but appears because the system stays in non-equilibrium. Hence, we observe the emergence of cyclic dominance as a chance event only. We have also simulated a model with rescaling of payoff so that the mean payoff becomes zero. In this case the overall population size is controlled by the control parameter M with \alpha=0. Diversity is only observed for large population sizes (large M values), implying that the diversity originated from a large mutation supply. Cyclic dominance does not contribute to diversity, and thus the underlying mechanism found based on the matrix approach also holds. We have cited the paper Scheffer and van Nes, 2006, and have discussed the results with the rescaling method in the Discussion.

**Author response image 1. respfig1:** The average fraction \chi of cyclic dominance compared to non-cyclic dominance in time. The payoffs are rescaled every mutation and extinction events so that the average of payoffs becomes zero. The population starts from a single-type population with the payoff zero. 2000 realizations are used for obtaining averages. At the steady-state, the average fractions are smaller than 0.02, which are smaller than that of our model.

In the same spirit:"For small α values (rich environments) in particular, the population size N at the steady state becomes large, containing many different types" the value of α should change nothing except the biomass scale (population size N ~ M lambda/alpha – incidentally, why introduce parameter M at all? it is never explained). The fact that more types exist for lower α seems like an artefact, perhaps because it takes longer for interactions to tend toward neutrality.

We agree that this deserves a better justification. We originally introduced the two parameters α and M for two different reasons: When the interaction parameters A_ij_ become large, α ensures that the population size remains restricted – and the magnitude α controls the importance of competition in the absence of any game theoretical interactions. On the other hand, M gives us the possibility to control the impact of the overall competition term. However, since we only focused a single M value, we have now dropped M by rescaling α and shifting A_ij_ by _lnM_. We now have used this alternative notion which hopefully reduces confusion, and thus parameters are recalculated while the results are robust.

8) Perhaps one could directly show how adding correlations in the matrix diminishes the occurrence of cyclical dominance? This would directly show that nothing else is needed (i.e. that the eco-evo process does nothing more than add these correlations).

In our model, the payoff correlations are an emergent property. The payoff matrix changes according to the simple rule with the correlation between parental and offspring types. But we do not impose any genealogies. Because the genealogy is an outcome of the eco-evo process, the whole structure of payoff correlations is also an outcome of the eco-evo process determining the fraction of emerging cyclic and non-cyclic dominance. Thus, introducing external correlations seems to come with a risk of introducing artefacts.

9) It is worth noting that there is a large ecological literature about extensions of cyclic dominance to more than triplets, called intransitive competition. Some of that literature has claimed (although we are not necessarily convinced) that intransitive competition is actually common and important. We would suggest reading these articles and figuring out why their claims would be so different – and if that difference is important, then discussing it in the article.See for instance as starting points: Soliveres et al., 2015; Gallien et al., 2017.

Thank you for these pointers. First of all, we would like to say that the findings in those papers and in our manuscript are not mutually exclusive. In our model, the intransitive links always appear but it does not always connect to the emergence of cyclic triplets due to other link types such as coexistence and bistability. This was also observed in a soil bacterial community (Kotil and Vetsigian, 2018); the structure is hierarchical with some intransitive relationships, which do not lead to the cyclic dominance. Second, we considered non-equilibrium steady state while others have focused on equilibrium states. The cyclic dominance we studied stands out from other forms of intransitive competition since it is virtually unable to evolve from the equilibrium state. This a major difference may be that ecologists typically think about the interaction between different species and evolutionary biologists more in terms of the origin of the interacting partners. Reconciling both aspects, our manuscript supports the basic idea that assembly of unrelated types is more likely to lead to cyclic triplets than evolution, in which emerging types are closely related. We have addressed this literature in the Discussion.